# Optogenetic and chemogenetic approaches reveal differences in neuronal circuits that mediate initiation and maintenance of social interaction

**Karolina Rojek-Sito[1], Ksenia Meyza[1], Karolina Ziegart-Sadowska[1], Kinga Nazaruk[1], Alicja Puścian[1], Adam Hamed[2], Michał Kiełbiński[3], Wojciech Solecki[3], Ewelina Knapska**[1] *

**1** Laboratory of Emotions Neurobiology, BRAINCITY—Centre of Excellence for Neural Plasticity and Brain Disorders, Nencki Institute of Experimental Biology, Polish Academy of Sciences, Warsaw, Poland, **2** Laboratory of Spatial Memory, Nencki Institute of Experimental Biology, Polish Academy of Sciences, Warsaw, Poland, **3** Department of Neurobiology and Neuropsychology, Institute of Applied Psychology, Jagiellonian University, Krakow, Poland

* e.knapska@nencki.edu.pl

**Data Availability Statement:** All data files are available from the database: Mendeley Data: DOI: https://data.mendeley.com/datasets/h49vtpjm8f/3.

## Introduction

Social species, including humans, are motivated not only by physiological needs such as food, water, or reproduction but also by the need to interact with other individuals and form relationships. The neural circuits promoting social interactions remain poorly understood. Previous studies implicated the neural circuits within the mesocorticolimbic reward system, including the ventral tegmental area (VTA), nucleus accumbens (NAc), central amygdala (CeA), orbitofrontal cortex (OFC), and anterior cingulate cortex (ACC) in social interaction [1–11].

Here, we aimed at dissecting whether neural circuits governing the motivation to initiate social contact and to maintain social interaction overlap or are distinct. The initiation of social contact necessitates the capacity to recognize and interpret social cues and signals from others, a motivation to engage in social interactions, and the willingness to physically approach others. On the other hand, the maintenance of social contact relies on the ability to synchronize behaviors, actions, and emotions with others, the reciprocation of social behaviors between individuals, and sustained engagement.

To explore this problem, we directed our attention to specific brain regions implicated in motivation and approach behaviors, such as the VTA and CeA, as well as regions associated with social cognition and executive functions, such as the ACC and OFC. We sought to investigate the causal role of this neural circuitry, characterized by its long-range functional connectivity, in different aspects of social interaction using a rat model. We analyzed multiple dimensions of social interaction, including initiation (reflecting the ability to initiate social contact), maintenance (reflecting the capacity to appropriately respond when a partner initiates social contact), and blocking (representing attempts to avoid social interaction initiated by the partner). By examining these various aspects, we aimed to gain insights into the specific contributions of this circuitry to different facets of social behavior.

The VTA, OFC, and ACC have previously been associated with facilitating positive social interactions [2–8,10]. The role of the CeA in driving social interactions remains largely unexplored. Previous studies have linked different subpopulations of neurons within the CeA to the

**Funding:** This work was supported by the European Research Council Starting Grant (H 415148) to EK (supporting the work of KRS, KM, KZS, KN, AH, and EK) and the BRAINCITY - Centre of Excellence for Neural Plasticity and Brain Disorders' project of the Foundation for Polish Science to EK (supporting the work of KRS, AP, and EK). The funders had no role in study design, data collection and analysis, decision to publish, or preparation of the manuscript.

**Competing interests:** The authors have declared that no competing interests exist.

**Abbreviations:** ACC, anterior cingulate cortex; AP, anteroposterior; ASD, autism spectrum disorder; CeA, central amygdala; DA, dopamine; DOPAC, dihydroxyphenylacetic acid; DV, dorsoventral; HVA, homovanillic acid; IP, intraperitoneally; ML, mediolateral; NA, noradrenaline; NAc, nucleus accumbens; NGS, normal goat serum; OFC, orbitofrontal cortex; PR, Progressive Ratio; USV, ultrasonic vocalization; TH, tyrosine hydroxylase; VTA, ventral tegmental area; 3-MT, 3-methoxytyramine.

modulation of rewarding experiences and the pursuit of food rewards [12–17]. Furthermore, it has been also demonstrated that oxytocin signaling in the CeA modulates the discrimination of emotions during social interactions [11,18], which is crucial for responding appropriately to the behavior of social interaction partners. The CeA exhibits strong connections with other structures in the mesocorticolimbic circuitry, and its stimulation recruits the VTA, NAcc, and OFC [12,19]. However, the functional distinction of CeA circuits based on their connectivity to different upstream and downstream brain areas remains unknown.

Using opsins selectively expressed in the neurons activated by social interaction, projection-specific chemogenetic manipulations, and functional tracing, we demonstrate that the neural circuit composed of neurons in the CeA, VTA, ACC, and OFC is critical for promoting social interactions. In particular, some projections within the circuit affect the initiation of social contact (VTA-ACC, VTA-OFC, OFC-CeA), while others are critical for maintaining it (CeA-VTA, ACC-CeA). Presented results provide novel insights into the social brain and lay a foundation for developing therapeutic approaches targeted at specific aspects of social interaction deficits.

## Results

### Optogenetic activation of the CeA cells increases social contact

To induce a robust drive for social interaction, we separated male rat cagemates [20,21] for 3 weeks. In rats, such procedure induces strong social motivation. Indeed, when the rats got reunited, they engaged in intensive positive interactions without signs of agonistic behaviors. Consistently, the rats produced positive 50-kHz ultrasonic vocalizations (USVs), but they did not produce aversive 22-kHz USVs (Figs 1A, 1B, and S1A). We found that interaction with a partner activates some CeA neurons as measured by the c-Fos expression (henceforth, we will call them "social cells"; Figs 1C, 1D, and S1B). To verify the role of the c-Fos–expressing neurons in social interaction, we used the behaviorally driven *c-fos*–dependent expression of channelrhodopsin [18]. We injected the AAV-c-fos-ChR2 virus and implanted optic fibers into the CeA of the experimental rats. Next, as previously described, we separated the animals for 3 weeks upon which the animals were reunited, which induced the *c-fos*–dependent expression of the ChR2 in social cells. To reactivate the neurons recruited by the social interaction, we stimulated them optogenetically on the next day in the presence of the partner rat (Fig 1E–1H). In the control group, the rats were treated the same way except for laser being off during the social interaction on the second day. Increase in the activity of social cells caused by photostimulation caused a marked increase in social contact, indicating that the targeted population of the CeA cells is causally involved in social interaction.

To further test the role of the CeA social cells and their behavioral specificity, we conducted another experiment to examine their involvement in motivation for food reward. Previous work has shown that optogenetic excitation of the CeA amplifies motivation to pursue food rewards in the progressive ratio lever pressing test [15]. Thus, we expected that if the social and food cells in the CeA overlap, their activation or inhibition will also affect food motivation. To test this, we induced the activity-dependent expression of opsins (specifically, the c-fos-ChR2 construct described above) in 2 groups of rats using the following methods: (a) social interaction (social cells); and (b) lever pressing for food (referred to as food cells; Figs 1I–1K and S1C). Then, we tested the effects of their photoactivation on motivation for food in the progressive ratio test. Consistently with the previous reports, we found that photoactivation of the food cells increased motivation for food, as measured by lever pressing. At the same time, activation of the social cells decreased it (Fig 1K). As the photoactivation of the social cells

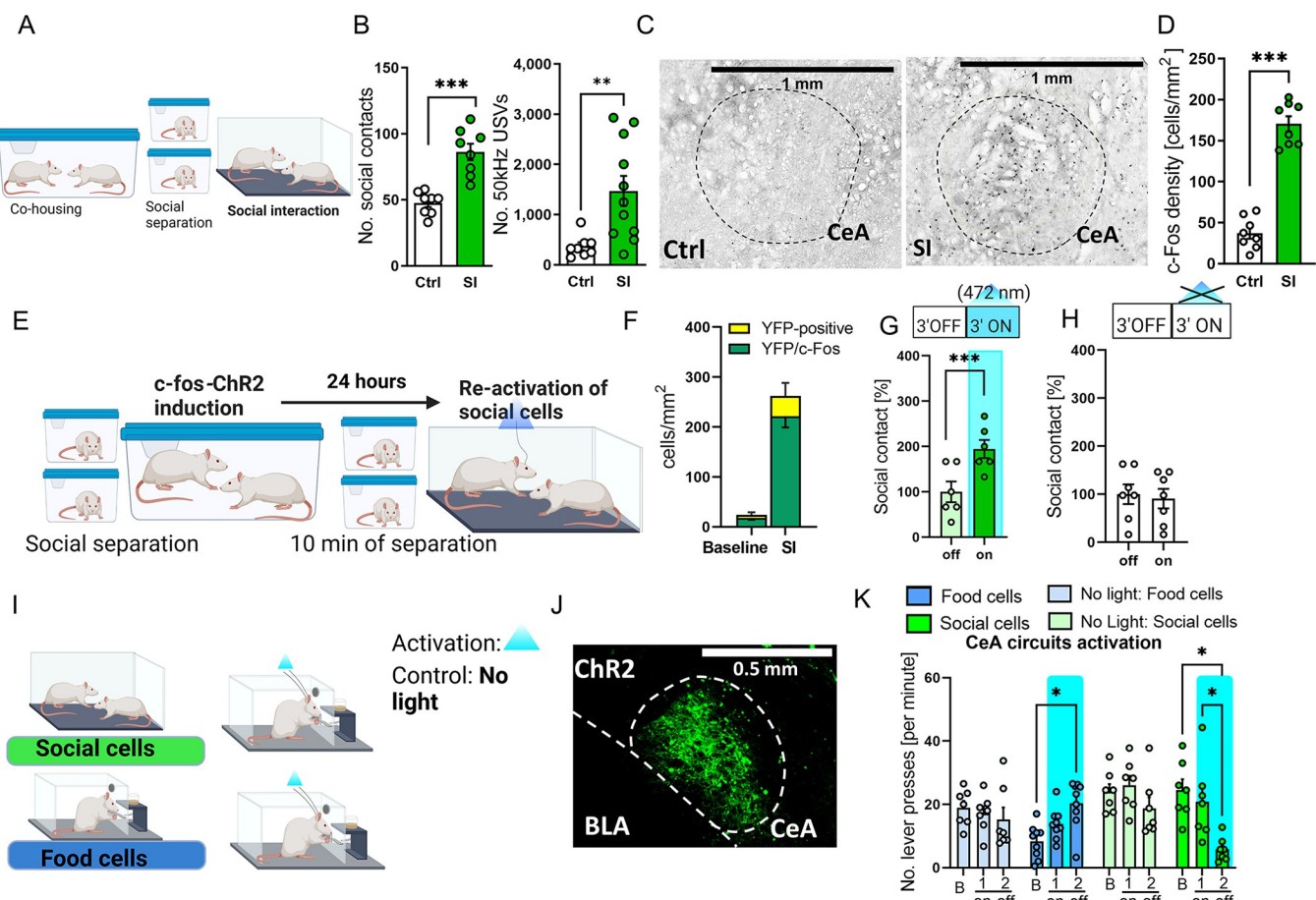

**Fig 1. Social interaction involves CeA neurons. (A)** The schematic of the experiment. Cagemate rats were separated for 3 weeks and then subjected to the social interaction (SI group). The control group had the social interaction after a brief, 10-minute separation (Ctrl). **(B)** The SI rats engage in intensive positive interaction. Number of social contacts and 50 kHz (positive) USVs during the 10-minute social interaction (social contacts: SI, $n = 8$; Ctrl, $n = 8$; unpaired $t$ test: t(14) = 5.588, $p < 0.0001$; USVs: SI, $n = 11$; Ctrl, $n = 8$; unpaired $t$ test: t(17) = 3.076, $p = 0.0068$). **(C)** Interaction with a partner activates the CeA cells. Representative images of c-Fos expression in the CeA in the SI and Ctrl groups. **(D)** Quantified c-Fos expression in the CeA (SI, $n = 8$; Ctrl, $n = 8$; unpaired $t$ test: t(14) = 11.63, $p < 0.0001$). **(E)** The schematic of the photomanipulation experiments—reactivation of social cells during social interaction. **(F)** Social interaction in the SI group induces strong expression of the optogenetic construct, measured as the YFP fluorescence colocalizing with the endogenous c-Fos (Social cells, $n = 3$). The baseline expression measured in animals with no exposure to social interaction after 3 weeks of social separation (Ctrl, $n = 3$). **(G)** Photoactivation of the CeA cells activated in the SI group increases social contact. Percentage of social contacts during the optogenetic reactivation of the social cells ("light on", 3-minute period) and without stimulation ("light off", 3-minute period) normalized to the Ctrl group; paired $t$ test (t(5) = 17, $p < 0.001$), $n = 6$. **(H)** There is no difference in the control group without laser stimulation ("on" 3-minute, "off" 3-minute, $n = 7$). **(I)** The schematic of the photomanipulation experiments—reactivation of social cells and food cells during food motivation test. **(J)** Representative images of the c-fos–dependent expression of the excitatory ChR2-YFP opsin in neurons activated by the social interaction (social cells). **(K)** Activation of the food cells increases lever pressing, while activation of the social cells decreases lever pressing (compared to baseline); controls show no difference between the baseline and following phases; two-way ANOVA (time × group effect: F(6,52) = 8.052, $p < 0.0001$), followed by Holm–Sidak post hoc tests. The average number of lever presses per minute during the baseline period (2 minutes), and ON–OFF (3 minutes) laser periods are shown. Ctrl: Social cells: $n = 7$, Ctrl: Food cells: $n = 7$, Activation: Social cells: $n = 7$, Activation: Food cells: $n = 9$. CeA, central amygdala. All the data are shown as the mean ± SEM, and dots represent individual data points, * $p < 0.05$, ** $p < 0.01$, *** $p < 0.001$. The data underlying this figure can be found in https://data.mendeley.com/datasets/h49vtpjm8f/3.

promotes social interaction with the partner, this suggests that populations of social and food cells do not overlap completely.

Then, using the c-fos-NpHR construct, we expressed halorhodopsin in neurons that were activated by either social interaction or lever pressing for food. We repeated the experiment described above, this time photoinhibiting either the social cells or the food cells. We found that inhibiting both food and social cells reduces motivation for the food reward and decreases cage exploration, suggesting some functional overlap between these populations (S1D–S1G

Fig). Notably, the behavior during sessions of social interaction or lever pressing for food used for induction of *c-fos*–dependent constructs did not differ between the groups (S1H–S1K Fig). All the procedures were the same in the control groups, except that we did not use laser light during the test. The number of social and food cells activated in the CeA was comparable (S1J and S1K Fig).

The photomanipulation experiments suggested that there is some functional overlap between the social and food cells, but it is not complete. We did not observe anatomical segregation between the social and food cells. Therefore, to investigate the factors that differentiate the social and food cells, we focused on identifying the CeA outputs of these cell populations. To that end, we measured the level of neurotransmitters and their metabolites in various brain areas after manipulating the activity of the social and food cells to identify the brain regions functionally connected with the CeA social but not food cells (S2 Fig). Comparing the levels of neurotransmitters in different brain structures after the stimulation of food and social cells, we observed changes in neurotransmitter release in the VTA, ACC, OFC, PFC, and NAc following the activation of social cells but not food cells (S2 Fig). We directed our focus towards the CeA-VTA-ACC/OFC pathways. This emphasis stemmed from the tracing experiment we conducted, which demonstrated high activity of the CeA-VTA projection during social interactions (see below). Furthermore, the changes observed in the VTA led us to speculate that the dopaminergic projection could be involved. Accordingly, we observed changes in dopamine levels in the ACC.

## The CeA-VTA projection is critical for the maintenance of social interaction

First, to investigate whether the CeA-VTA projection is activated by social interaction, we used c-fos-PSD95-Venus rats, which express a reporter protein under the control of the *c-fos* promoter, and injected them with an anterograde tracer (PHA-L: Phytohemagglutinin-L) into the CeA. Our findings revealed a dense projection from the CeA to the VTA cells activated during social interaction (S3A Fig).

Then, we performed the loss-of-function experiments on the CeA-VTA pathway. Using DIO-hM4Di and Cav-Cre constructs activated by the DREADD agonist c21 (Compound 21), we inhibited the CeA-VTA projection during social interaction (Figs 2A–2C, S3B, and S3C). We performed 3 types of comparisons: with the control group, with a partner (who was treated with saline), and with the baseline (after a short-time separation). Our primary focus for drawing conclusions was the comparison with the control group and analyzing the changes in partners' behavior. Furthermore, we measured attempts to block the interaction as they represent the negative aspects of maintenance. The comparison with the baseline was conducted to monitor any potential alterations in sociability that may have arisen during the experimental manipulations. The results of these analyses can be found in S3 and S5 Figs.

We discovered that inhibiting the CeA-VTA pathway disrupts the maintenance of social interaction, specifically affecting positive reactions to the partner's approach, such as sniffing or allogrooming, and sustained close physical contact. Inhibiting this pathway leads to an increase in active blocking of social contact. At the same time, rats with inhibited CeA-VTA pathways initiate social interaction, approaching their partners as frequently as animals in the control group (Figs 2D–2G, S3D, and S3E). Interestingly, the inhibition of the CeA-VTA pathway in one rat also alters the behavior of their partners, who attempt to initiate social contact more frequently than the control animals (Fig 2D). Despite the increased initiation of social contacts by the partners, rats with inhibited CeA-VTA pathways do not respond to these attempts and actively avoid social interaction. The manipulation did not affect rats' general

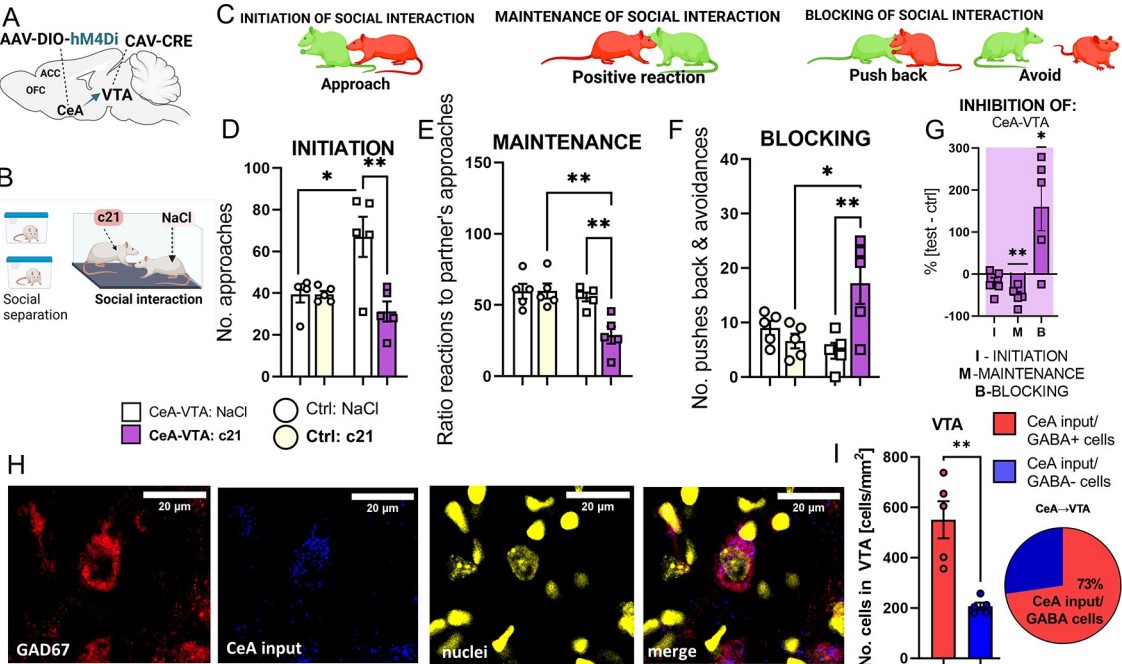

**Fig 2. Inhibition of the CeA-VTA projections disrupts the maintenance of social contact.** (**A**) Experimental protocol: Each of the co-housed pairs was subjected to inhibition of the CeA-VTA projection in one animal, preceded by the injections of the viral vectors: AAV-DIO-H4 and Cav-CRE. The control (Ctrl) animals were injected with the AAV-DIO-mCherry and Cav-CRE delivering viruses. (**B**) Assessment of social interaction in each subject (red rat) involved Left: Initiation of social interaction: approaching a partner; Middle: Maintenance of social interaction: percentage of the positive reactions to partner's approaches; and Right: Blocking of social interaction: pushing back or avoiding social interaction (see Methods for more details). (**C**) The schematic of the experiment. Co-housed animals were separated for 3 weeks and then subjected to the social interaction test. Prior to the test (30 minutes), one of the rats received an IP injection with the chemogenetic activator (c21), and the other with NaCl. (**D–F**) Inhibition of the CeA-VTA inputs disturbs the maintenance of social contact and leads to active blocking of social contact, but it does not affect the initiation of social contact, compared to the control group. Animals injected with c21 were compared with animals from the control group also injected with c21, as well as conspecific partners (white bars) injected with saline. It is worth noting the increase in social contact initiation by partners of the CeA-VTA:c21 rats. Initiation: one-way ANOVA (group effect: $F_{(3,16)} = 7.364$, $p = 0.0026$) followed by Holm–Sidak post hoc tests; Maintenance: one-way ANOVA (group effect: $F_{(3,16)} = 8.848$, $p = 0.0011$) followed by Holm–Sidak post hoc tests; Blocking: one-way ANOVA (group effect: $F_{(3,16)} = 6.018$, $p < 0.0060$) followed by Holm–Sidak post hoc tests; Ctrl: NaCl/c21 $n = 5/5$, CeA-VTA: NaCl/c21 $n = 5/5$. (**G**) The change in the initiation (I), maintenance (M), and blocking (B) of social interaction, computed as a percentage of the average results of the control group, during inhibition of the CeA-VTA projection yielded the following significant results (comparison to no change level, one-sample $t$ test): M: CeA-VTA: $p = 0.0064$ ($t = 5.221$, df = 4), B: CeA-VTA: $p = 0.0484$ ($t = 5.221$, df = 4); CeA-VTA:c21 $n = 5$, Ctrl:c21 $n = 5$. (**H**) Representative images of (from the left): the GABA-positive cells (anti-GAD67 ab) in the VTA (red), projections from the CeA (blue), nuclei of cells (yellow), and the merged image of all the former. (**I**) Left: Number of the GABA-positive cells in the VTA receiving projections from the CeA (red) and the number of the Gad67-negative cells receiving projections from the CeA (blue). Right: Majority of the CeA-VTA projections are onto the GABA + cells; paired $t$ test $t(4) = 4.028$, $p = 0.0158$; CeA-VTA: $n = 5$. All the data are shown as the mean ± SEM, and symbols represent individual data points, * $p < 0.05$, ** $p < 0.01$. The data underlying this figure can be found in https://data.mendeley.com/datasets/h49vtpjm8f/3.

locomotor activity (S3F Fig). The methods of tracing the CeA-VTA projection we used could also mark the collaterals of the CeA neurons projecting to the brain structures other than the VTA. Thus, to control for the collateral projections of the CeA neurons, we carefully inspected the brains of the injected animals.

Next, we examined the colocalization of CeA inputs to the VTA with GAD67, a marker of GABAergic neurons. The analysis revealed a high degree of colocalization, indicating that the cells in the VTA receiving projections from the CeA are predominantly GABAergic (Fig 2H and 2I). Further, activation of the CeA social cells led to a decrease of GABA in the VTA (S2 Fig). Together, these results suggest that the CeA output to the VTA may inhibit the tonic

inhibition exerted by GABAergic interneurons on dopaminergic neurons. If this hypothesis holds true, the CeA-VTA projection should increase the activity of dopaminergic neurons in the VTA. In our next experiment, we investigated the role of dopaminergic projections from the VTA to the ACC and OFC in social interaction.

## The dopaminergic VTA-ACC and VTA-OFC projections regulate initiation of social interaction

Firstly, using Th-Cre rats, we confirmed that the VTA sends dopaminergic projections to the ACC and OFC (Fig 3A). Then, we injected frt-hM3D(gq) into the VTA and CavFlexFlp into the ACC or OFC of Th-Cre rats, allowing for stimulating of the dopaminergic projections with the DREADD agonist C21 (Figs 3B–3D, S3G, and S3H). We found that the activation of the VTA-ACC and VTA-OFC had a significant effect on the initiation of social contact but not on

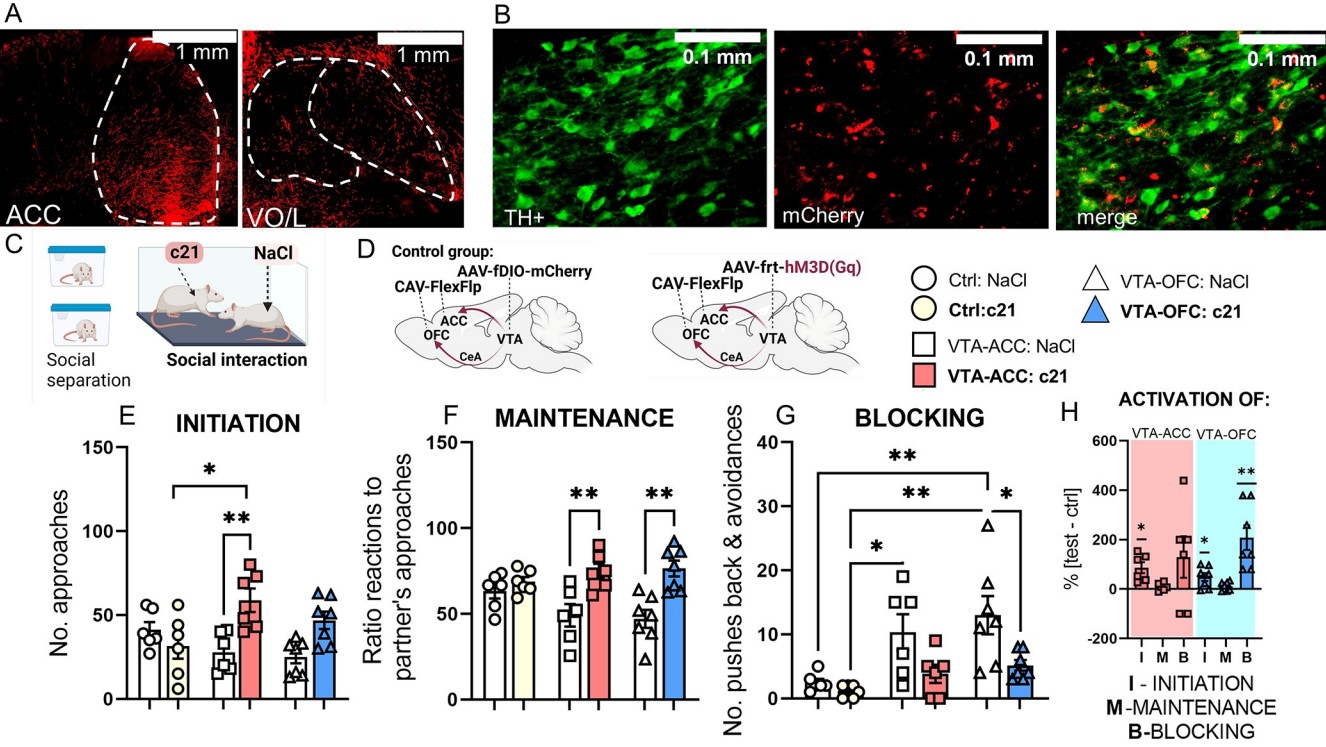

**Fig 3. Activation of the dopaminergic VTA-ACC and VTA-OFC projections increases social contact initiation.** (A) The representative images of the VTA inputs into the ACC (left) and OFC (right) in Th-Cre rats labeled with AAV-DIO-mCherry. The white lines mark the areas of the ACC and OFC. (B) Representative images of (from the left) the TH-positive cells (anti-Th ab) in the VTA (green), expression of AAV-hSyn-frt-hM3D(Gq):mCherry (red), and the merged image of all the former. (C) The schematic of the experiment. (D) Schematic of the chemogenetic activation of the VTA-ACC and VTA-OFC projections. (E, F) Activation of the VTA-ACC inputs enhances social contact initiation, while activation of neither the VTA-OFC nor VTA-ACC affects contact maintenance. Instead, activation of both VTA-ACC and VTA-OFC decreases maintenance of social contact by the partners of the treated animals. (G) Activation of both VTA-ACC and VTA-OFC pathways also results in active blocking of social contact by the partner rats. Animals injected with c21 were compared with animals from the control group also injected with c21, as well as conspecific partners (white bars) injected with saline. Initiation: one-way ANOVA (group effect: F(5,32) = 5.362, $p$ = 0.0011) followed by Holm–Sidak post hoc tests; Maintenance: one-way ANOVA (group effect: F(5,32) = 7.1897, $p$ = 0.0001) followed by Holm–Sidak post hoc tests; Blocking: one-way ANOVA (group effect: F(5,32) = 6.295, $p$ = 0.0004) followed by Holm–Sidak post hoc tests; Ctrl: NaCl/c21 $n$ = 6/6, VTA-ACC: NaCl/c21 $n$ = 6/6, VTA-OFC: NaCl/c21 $n$ = 7/7. (H) The change in the initiation (I), maintenance (M), and blocking (B) of social interaction, computed as a percentage of the average results of the control group, during inhibition of the VTA-ACC or VTA-OFC projections yielded the following significant results (comparison to no change level, one-sample $t$ test): I: VTA-ACC: $p$ = 0.0114 (t = 3,898, df = 5), I: VTA-OFC: $p$ = 0.0253 (t = 2,960, df = 6), B: VTA-OFC: $p$ = 0.0058 (t = 4,190, df = 6). Ctrl:c21 $n$ = 6, VTA-ACC:c21 $n$ = 6, VTA-OFC:c21 $n$ = 7. Initiation of social interaction: approaching a partner; Maintenance of social interaction: percentage of the positive reactions to partner's approaches; Blocking of social interaction: pushing back or avoiding social interaction. All the data are shown as the mean ± SEM, and symbols represent individual data points, $^*$ $p$ < 0.05, $^{**}$ $p$ < 0.01. The data underlying this figure can be found in https://data.mendeley.com/datasets/h49vtpjm8f/3.

its maintenance (Figs 3E–3H and S3I). Stimulation of the VTA-ACC and VTA-OFC pathways increased active approaches toward the partners. However, simultaneously, rats with activated VTA-OFC projections more frequently blocked their partner's attempts to interact compared to control animals (Fig 3G and 3H). These oversocial and inconsistent behaviors during the activation of the VTA-ACC/OFC led to a decrease in positive responses to contact initiations and an increase in avoidance of social interaction in the partner rats (Figs 3F–3H and S3I). Activation of neither the VTA-ACC nor VTA-OFC affected general locomotor activity (S3J Fig).

Finally, to test the specificity of the involvement of the CeA-VTA-cortical circuits in social interaction, we employed the food motivation test (S4 Fig). We tested well-trained animals lever pressing for food under a progressive ratio schedule of reinforcement that reflects their motivation to obtain the reward. We found that inhibition of the CeA-VTA projection, as well as activation of the VTA-ACC/OFC dopaminergic pathways had no effect on motivation for food reward, thus confirming that the observed effects were specific to social interaction.

## Inhibition of the OFC-CeA pathway disrupts primarily the initiation of social interaction, while inhibition of the ACC-CeA pathway affects its maintenance

Next, we focused on the projections from the ACC and OFC to the social cells in the CeA. To identify the social cells inputs, we injected the c-fos-PSD95-Venus rats, which express a reporter protein under the control of the *c-fos* promoter, with an anterograde tracer PHA-L to the ACC or OFC (Fig 4A). We induced expression of the reporter protein in the CeA by social interaction and then imaged the activated neurons counting how many of them receive inputs from the ACC or OFC (Figs 4B–4D and S5A). We confirmed that ACC and OFC neurons project onto the social cells in the CeA. Notably, we discovered that the ACC innervates more of the CeA social cells than the OFC.

To test the role of the ACC-CeA and OFC-CeA projections in social interaction, we inhibited either the ACC-CeA or OFC-CeA projections chemogenetically using DIO-hM4Di and Cav-Cre constructs activated by the DREADD agonist C21. We observed that the vast majority of the terminals were located in the places injected with Cav-CRE, confirming the reliability of the chosen method for the purpose of selective projection manipulation (Figs 4E–4G, S5B, and S5C). Behavioral analysis showed that the ACC-CeA and OFC-CeA differentially control various aspects of social interaction (Figs 4H–4K, S5D, and S5E). Specifically, we discovered that inhibition of the OFC-CeA pathway primarily impacts the initiation of social interaction. In contrast, inhibition of the ACC-CeA pathway predominantly affects the maintenance of social interaction when compared to the control group (Figs 4H, 4I, 4K, and S5E). Notably, when either the OFC-CeA or ACC-CeA pathways are inhibited, we observe an increase in attempts by the treated rats to block social interaction. This is evidenced by contact avoidance and pushing the partner away (Fig 4J and 4K and S1 and S2 Videos). The increased blocking behavior and decreased maintenance of social interaction in rats with an inhibited OFC-CeA pathway, compared to the partner rats, suggest additional challenges in maintaining social contact in the OFC-CeA group. Inhibition of neither the ACC-CeA nor OFC-CeA changed general locomotor activity (S5F Fig).

Finally, we tested the role of the ACC-CeA and OFC-CeA pathways in food motivation using progressive ratio test as described above. We found that inhibition of both projections decreased the number of operant responses for food reward, which contrasts with the effects of manipulating the CeA-VTA-ACC/OFC projections (S5G Fig).

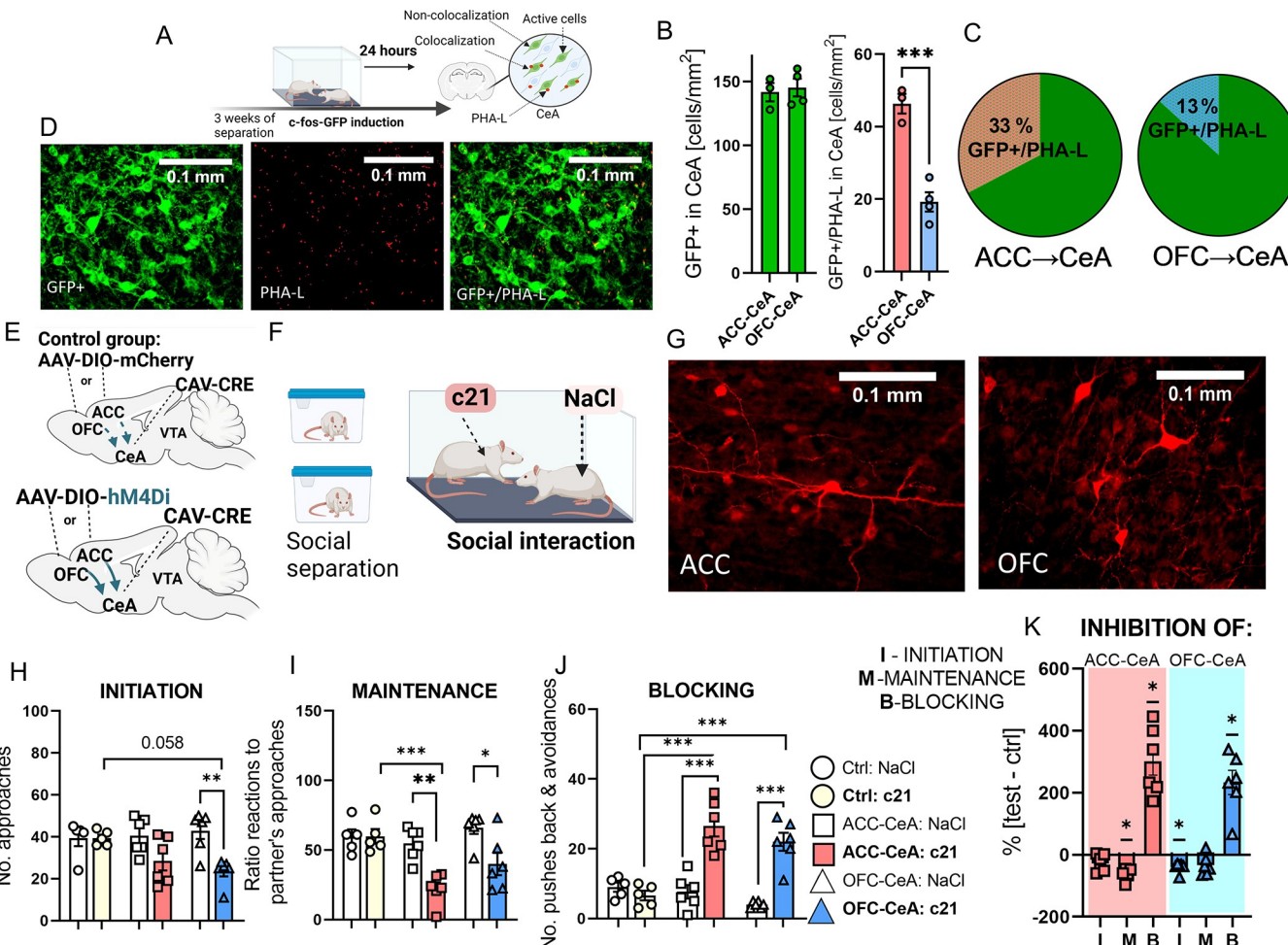

**Fig 4. Inhibition of the OFC-CeA pathway disrupts the initiation of social interaction, while inhibition of the ACC-CeA pathway impairs the maintenance of social interaction.** (**A**) The schematic of the experiment. Co-housed c-fos-PSD95-Venus rats were injected with anterograde tracer into ACC or OFC, separated for 3 weeks, and then subjected to the social interaction test. (**B**) Number of the CeA neurons that are activated by the social interaction (left), and a number of neurons that receive projections from the ACC or OFC (right); unpaired $t$ test (t(5) = 6.917, $p$ = 0.001, ACC-CeA: $n$ = 3, OFC-CeA: $n$ = 4. (**C**) Percent of the CeA neurons activated by the social interaction (green) that receive projections from the ACC (left, red), and OFC (right, blue), respectively. (**D**) Representative image of the c-fos–driven expression of the GFP in the neurons activated by the social interaction. Middle: anterograde tracer (PHA-L)-labeled projections from the ACC to CeA. Right: overlay of the left and middle images. (**E**) Schematic of the chemogenetic inhibition of the ACC-CeA, or OFC-CeA projections. (**F**) The schematic of the experiment. (**G**) The representative images of the AAV-hSyn-DIO-{hCAR}off-{hM4Di-mCherry}on-W3SL expression in the ACC and OFC. (**H, I**) Inhibition of the ACC-CeA projection disrupts the maintenance of social contact, while inhibition of the OFC-CeA projection impairs the initiation of social contact. (**J**) Inhibition of both pathways results in active blocking of social contact. Animals injected with c21 were compared with animals from the control group also injected with c21, as well as conspecific partners (white bars) injected with saline. Initiation: one-way ANOVA (group effect: F(5,28) = 4.941, $p$ = 0.0023) followed by Holm–Sidak post hoc tests; Maintenance: one-way ANOVA (group effect: F(5,28) = 8,960, $p$ < 0.0001) followed by Holm–Sidak post hoc tests; Blocking: one-way ANOVA (group effect: F(5,28) = 21.65, $p$ < 0.0001) followed by Holm–Sidak post hoc tests; Ctrl: NaCl/c21 $n$ = 5/5, ACC-CeA: NaCl/c21 $n$ = 6/6, OFC-CeA: NaCl/c21 $n$ = 6/6. (**K**) The change in the initiation (I), maintenance (M), and blocking (B) of social interaction, computed as a percentage of the average results of the control group, during inhibition of the ACC-CeA or OFC-CeA projections yielded the following significant results (comparison to no change level, Wilcoxon signed rank test): I: OFC-CeA: $p$ = 0.0313, M: ACC-CeA: $p$ = 0.0313, B: ACC-CeA: $p$ = 0.0313, B: OFC-CeA: $p$ = 0.0313; ACC-CeA:c21 $n$ = 6, OFC-CeA:c21 $n$ = 6, Ctrl:c21 $n$ = 5. Initiation of social interaction: approaching a partner; Maintenance of social interaction: percentage of the positive reactions to partner's approaches; Blocking of social interaction: pushing back or avoiding social interaction. All the data are shown as the mean ± SEM, and symbols represent individual data points, * $p$ < 0.05, ** $p$ < 0.01, *** $p$ < 0.001. The data underlying this figure can be found in https://data.mendeley.com/datasets/h49vtpjm8f/3.

## Discussion

In this study, we conducted specific neuronal manipulations to investigate the role of the CeA-VTA-ACC/OFC-CeA circuit in social interaction. Successful social interaction relies on

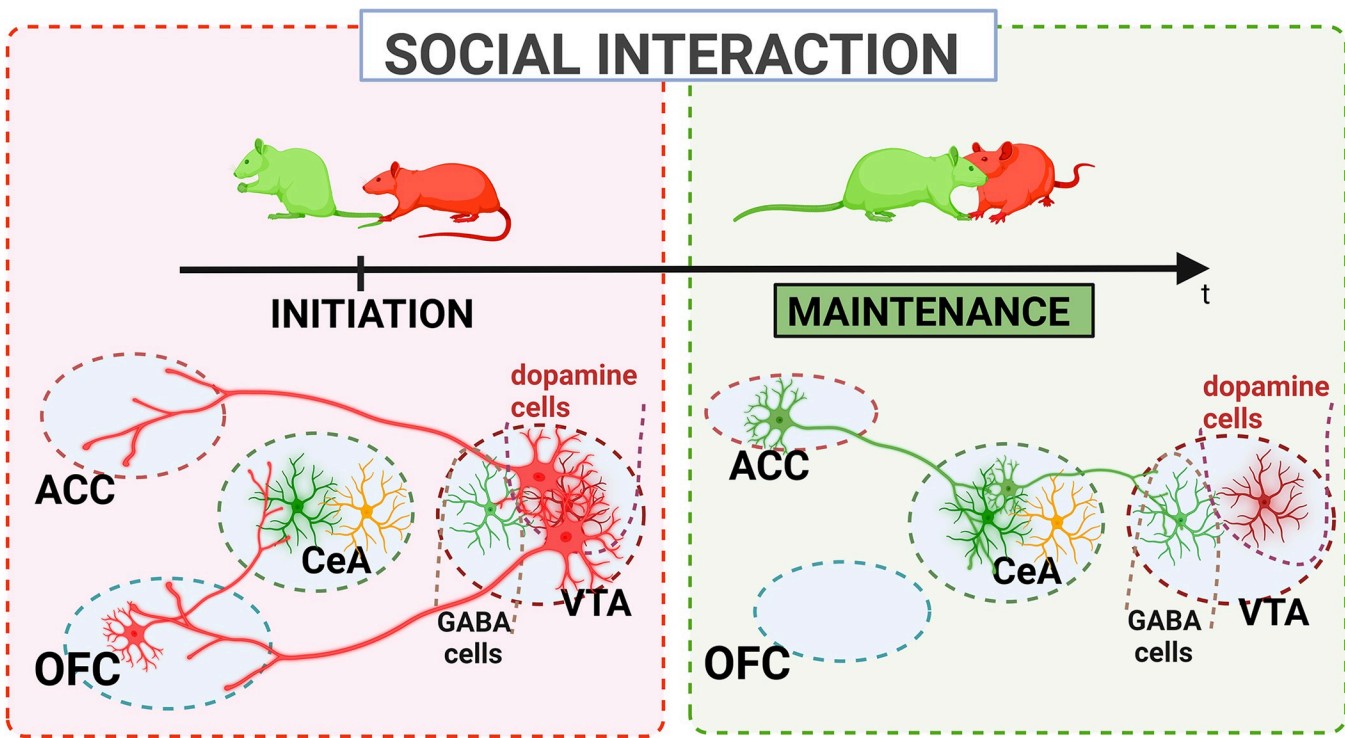

**Fig 5. Graphical summary of the ACC/OFC-CeA-VTA-ACC/OFC projection's role in the initiation of social interaction and its role in the maintenance of social interaction.**

2 fundamental conditions. Firstly, individuals must display the motivation to initiate interaction and an ability to recognize and interpret social cues and signals from others in order to initiate social contact. Secondly, they need the skills and capability to sustain the interaction for a time sufficient to synchronize behaviors, actions, and emotions with others. Consequently, when analyzing the behavioral outcomes of the manipulation of such circuit activity, we comprehensively examined various aspects of social interaction, encompassing both its initiation and maintenance. By doing so, we aimed to gain insights into the distinct neural circuits that underlie the diverse skills required for initiating and maintaining social interaction. Furthermore, we classified avoidance of social interaction and pushing back the partner rat as a third category of social interaction, referred to as "blocking." This category represents the unwillingness or aversion to engage in social contact and can be considered an attempt to prevent both the initiation of social interaction by others and its maintenance.

Our study provides evidence of the impact of manipulating the OFC-CeA, VTA-ACC, and VTA-OFC projections on the initiation of social contact. In contrast, manipulation of the ACC-CeA and CeA-VTA projections primarily affects responses to partners' attempts to initiate social interaction. This observation highlights the functional specificity of these circuits in the social domain (Fig 5).

To further investigate the circuits' involvement in social interaction, we conducted a comparative analysis with food motivation. Our aim was to determine whether these circuits play a general role in motivation or if their function is specific to social interaction. Surprisingly, our findings reveal that manipulating the CeA-VTA-ACC/OFC projections has no discernible impact on food motivation. However, it exerts a specific influence on social interaction. These results underline the specialized nature of these circuits in the social brain.

Social interactions are inherently rewarding experiences that elicit approach behaviors towards other individuals [22,23]. Previous research has implicated several brain structures, including the VTA, OFC, and ACC, which are all part of the reward system, in mediating positive social interactions [2–7,10,17]. However, the specific connectivity between these brain areas and other components of the circuit involved in social interactions remained largely unknown. In our study, we discovered the crucial role of neuronal circuits within the CeA in mediating social interaction. These circuits can be characterized by their inputs from the OFC and ACC, as well as their outputs to the VTA, OFC, and ACC.

The CeA is part of the reward system, and its role in processing food and drug rewards is well established [14]. Previous research has shown that optogenetic activation of the CeA amplifies motivation to pursue food and drug rewards [15,24] or can induce maladaptive attraction to the painful stimuli [12]. Additionally, recent work has implicated the CeA in processing information about the affective states of others [11,18]. Here, we show that photoreactivation of the CeA neurons previously engaged in positive social interaction increases social contact. Additionally, further specification of the CeA social population shows that reactivation of these cells suppresses motivation for pursuing food, in contrast to the reactivation of the CeA cells previously engaged in food reward. Suppression of food motivation during activation of social cells in the CeA suggests that social cells block food cells, promoting social contact over food. Together, these results show that the populations of social and food cells are, at least partially, functionally distinct. In line, we find that stimulation of social but not food cells in the CeA modulates neurotransmitter release in the VTA, OFC, and ACC. We further confirmed the involvement of these projections specifically in social interaction in functional studies.

The amygdala has long been considered a part of the "social brain," and its abnormality has been linked to deficits in social behavior [25]. However, the structure is heterogeneous, comprising the cortex-like basolateral part and striatum-like part, including the CeA. Previous studies showed that the projections from the basolateral part inhibit social interaction [26,27]. Less is known about which circuits promote social contact. We show that the CeA, which mediates reward motivation, plays such a role. Recently, the medial amygdala–hypothalamus circuit has also been implicated in social reward [28]. As the CeA receives projections from the medial amygdala [29], it is feasible it integrates information from the cortex and medial amygdala in this context.

The VTA, a key source of dopamine in the cortex, plays a crucial role in encoding motivation for social interactions [30,31]. It is a heterogeneous region composed of various cell types, including neurons that release dopamine (DA), GABA, or glutamate [31,32]. The activity of DA neurons is regulated by GABAergic interneurons [31]. Our findings demonstrate that the CeA-VTA projection, which primarily targets GABAergic neurons in the VTA, mediates the maintenance of social interaction, likely through disinhibition of DA neurons in the VTA. However, further studies are necessary to elucidate the interactions between CeA projecting neurons and VTA GABAergic and DA neurons that drive social behavior.

Previous studies have demonstrated that the processing of social stimuli involves dopamine receptors [33] in the ACC and OFC [34]. In particular, recent research suggests that the dopamine system in the OFC may play a crucial role in the development of social anxiety disorder [34]. Furthermore, studies have indicated that the participation of dopamine D2 receptors in the ACC is essential for processing social information during social observational fear learning in animals [33]. Consistent with these findings, the specific stimulation of dopaminergic VTA-ACC or VTA-OFC projections resulted in an increased initiation of social interaction, highlighting the critical role of these projections in modulating social behavior. Interestingly, this heightened sociability persisted despite the partners' attempts to avoid social contact,

indicating potential challenges in recognizing and interpreting social signals from others in rats with overactive VTA-ACC/OFC pathways.

Furthermore, our findings revealed that both the ACC and OFC cortical regions, known to be involved in social decision-making [35–39], project to CeA neurons that are activated during social interaction with a partner. Inhibition of the OFC-CeA projection disrupted the initiation of social interaction, while inhibition of the ACC-CeA pathway disturbed its maintenance. The anterograde transport tracers used to track the ACC-CeA and OFC-CeA projections were injected into the ACC and OFC of separate animals, thereby preventing a direct comparison. However, our findings indicate that the ACC innervates a greater number of social cells in the CeA compared to the OFC. This suggests the possibility that certain cells may receive innervation from the ACC but not the OFC. However, a systematic comparison is necessary to gather evidence regarding whether the same CeA neurons receive projections from both the ACC and OFC.

The specific neuronal circuits responsible for the initiation and maintenance of social interaction had not been previously investigated. Our results provide evidence that distinct neuronal circuitry mediates the different abilities required for initiating and maintaining social contact. Additionally, we demonstrate that manipulating neuronal projections involved in the initiation and/or maintenance of social interaction can impact not only the behavior of the treated animals but also the response of their interaction partners. This can manifest as the discontinuation of interaction and aversion to social contact, potentially indicating abnormalities in the behavior of the treated rats.

Most of the previous work on the CeA circuits focused on the neuronal populations identified by the molecular markers they express. This approach revealed reciprocally connected neuronal circuits that control specific, often opposite behaviors [13]. However, in contrast to the inputs and outputs, the markers usually do not define the function of the neuronal population. Notably, the marker-defined populations are often heterogeneous and can control many behaviors [13]. Thus, our study defined cell populations by their functional connectivity rather than the markers they express.

Understanding the specific circuits involved in initiating and maintaining social interaction can offer valuable insights into neurodevelopmental disorders such as autism spectrum disorder (ASD). The absence of social-seeking tendencies observed in individuals with ASD can be attributed to deficits in social motivation. Research has revealed that infants with a genetic predisposition to ASD exhibit diminished social motivation and interest. This deficiency fundamentally alters how individuals with ASD engage with and perceive the world, resulting in a lack of crucial opportunities to develop social perceptual and social cognitive abilities. Impaired social cognition and skills can thus be considered secondary effects stemming from reduced social motivation, potentially involving distinct neuronal circuits [40–42]. By addressing the core deficits in social motivation, there may be a possibility to enhance the development of social cognition and skills in individuals with ASD.

## Methods

### Animals

The study was conducted using Wistar rats supplied by the Center of Experimental Medicine in Bialystok, Poland; c-Fos-PSD95Venus-Arc rats (axonal tracing experiments) bred at the Nencki Institute Animal House and the Animal House at the Faculty of Biology, University of Warsaw, Poland; and Sprague Dawley tyrosine hydroxylase (TH) IRES-Cre rats+/− transgenic rats [43] (chemoactivation of dopaminergic pathways) supplied by the Institute of Zoology and Biomedical Research, Jagiellonian University (Krakow, Poland) under a breeding license

with Horizon Discovery (Vienna, Austria). Only male animals were used for all the experiments conducted. The rats weighed 130 to 150 g at the beginning of the experiment.

The animals were randomly paired and subsequently housed in pairs in standard home cages ($43.0 \times 25.0 \times 18.5$ cm) under a 12-hour light/dark cycle (lights on at 7 AM) at 22°C and controlled humidity (around 55%). All experiments were performed during the light phase of the cycle. A mild food restriction, achieved by providing 15 g of food per rat per day (standard laboratory chow), was applied to reduce and maintain rat weight at approximately 85% of their free-feeding weight. The food restriction began 7 days before the beginning of the training and was continued until the end of the experiment. Water was freely available. The rats were habituated to the experimenter's hand for 14 days preceding the experiment. This was followed by habituation to the transport, experimental rooms, and equipment (Skinner box and social interaction box) 3 days before the first training session. During social separation phase, all animals were singly housed (in standard cages, $26.5 \text{ cm} \times 42.5 \text{ cm} \times 18.5 \text{ cm}$) for 22 days. All experiments were carried out in accordance with the Polish Act on Animal Welfare after obtaining specific permission from the First Warsaw Ethical Committee on Animal Research.

## Viral vectors and tracers

For the optogenetic experiments, we used c-fos-ChR2/c-Fos-NpHR viral vectors. Our constructs were made as described earlier [13]. Following isolation using EndoFree Maxi Prep (Qiagen, S1 Table), the pAAV-c-Fos-ChR2(H134R)-EYFP and pAAV-c-fos-NpHR-EYFP plasmids were used to generate recombinant AAV vectors of mosaic serotype ½. Vectors were purified from cell lysates and their genomes were titrated by quantitative PCR.

For the anterograde tracing, the Venus rats were injected with anterograde axonal transport tracer PHA-L Alexa Fluor 647 conjugate (2.5% wt/vol, dissolved in 0.1 M PBS (pH 7.4); Invitrogen; S1 Table).

For chemoinhibition experiments, we used the pAAV-hSyn-DIO-{hCAR}off-{hM4Di-mCherry}on-W3SL plasmid with facilitating efficient retrograde spread in long-range projections purchased from Addgene [44]. Following isolation using EndoFree Maxi Prep, the pAAV-hSyn-DIO-{hCAR}off-{hM4Di-mCherry}on-W3SL plasmid was used to generate recombinant AAV vectors serotype 9. For control group, we used AAV9-hSyn-DIO-mCherry (AddGene; S1 Table). For initiated expression of receptors, we used Cav2-CreGFP (IGMM, Montpellier, France; S1 Table).

For chemoactivation experiments, we used the pAAV-hSyn-frt-hM3D(Gq):mCherry plasmid, a generous gift from Roger A. Adan, Utrecht University, the Netherlands [45]. Following isolation using EndoFree Maxi Prep, the pAAV-hSyn-frt-hM3D(Gq):mCherry plasmids were used to generate recombinant AAV vectors serotype 5. In control groups, we used AAV5-EF1a-fDio-mCherry (AddGene; S1 Table). For initiated expression of receptors, we used CAV-FLExloxP-Flp (CavFlexFlp, IGMM, Montpellier, France; S1 Table).

## Surgery

**For photomanipulation experiments.** Rats received intracranial injections of c-fos-ChR2 (400 nL/site) or c-fos-NpHRR3 (350 nL/site) viral vectors 3 weeks before the behavioral experiment. All surgical instruments were sterilized before surgery. Rats were anesthetized with isoflurane (5% induction, 2% for maintenance) and were given a subcutaneous injection of an analgesic (Butorfanol, Butomidor, 3 mg/kg). Ocular lubricant was used to moisten the eyes and the scalp was shaved. After being placed into the stereotaxic apparatus (David Kopf Instruments), the scalp was disinfected with 70% (vol/vol) alcohol, incised, and retracted. Two small

burr holes were drilled to allow for a 1-μL NanoFil syringe needle (World Precision Instruments) to be lowered into the desired part of the brain. All coordinates for stereotaxic surgeries were obtained from the rat brain atlas [46] with anteroposterior (AP), mediolateral (ML), and dorsoventral (DV) positions referenced from Bregma. The coordinates used for the CeA were as follows: AP, −1.8 mm; ML, ±3.8 mm; DV (from dura), −7.7 mm. The viral vector was delivered into the CeA (100 nL/minute) using UMP3 UltraMicroPump (World Precision Instruments), and the syringe remained in place for another 7 minutes to allow for the diffusion of the viral vector. Optic cannulas (200 μm in diameter) were implanted bilaterally 0.2 mm above the injection sites and afterwards secured with 2 skull screws and dental cement. The animals were administered an analgesic (Tolfenamic acid, Tolfedine; 4 mg/kg; SC) and an antibiotic (Enrofloxacin, Baytril 2.5 mg/kg; SC). To avoid dehydration, the animals were given 1 mL of 0.95% NaCl/100 g of body weight by SC injection. The rats were kept on a heating pad until they recovered from anesthesia before placing in a single cage. The behavioral experiment was conducted after 3 weeks of social separation started after the surgery.

**For the anterograde tracing experiments.** The surgeries were performed 21 days before the behavioral experiment. The PHA-L was injected into the ACC (AP), −1.8; (ML), ±0.8; (DV), −1.8, the OFC (AP), +3.7; (ML), ±2.1; (DV) −4.2, or to the CeA (AP), −1.8; (ML), ±3.8; (DV) −7.7. The procedure was identical as described above, with the exception that the optic cannulas were not implanted, so the incisions were sutured (no dental cement was used).

**For chemoinhibition experiments.** (ACC-CeA, OFC-CeA, CeA-VTA), the Wistar rats were injected with AAV9-hSyn-DIO-{hCAR}off-{hM4Di-mCherry}on-W/3SL or AAV9-h-Syn-DIO-mCherry (the control groups). The infusion was made into one of the cortical regions: to the ACC (AP), +1.6; (ML), ±0.8; (DV), −1.8, or to the OFC (AP), +3.7; (ML), ±2.1; (DV), −4.2, or to the CeA (AP) −1.8; (ML), ±3.8; (DV), −7.7. Three weeks later, rats were injected Cav2-CreGFP into CeA (AP), −1.8 mm; (ML), ±3.8 mm; (DV, from dura) −7.7 mm or into the the VTA: (AP), −5.35; (ML), ±1.1; (DV), −8.0. The behavioral experiment was conducted after 3 weeks of social separation started after the second surgery.

**For the activity of dopaminergic projections manipulation experiments.** Th-Cre rats were injected with the AAV5-hSyn-frt-hM3D(Gq):mCherry or with the AAV5-EF1a-fDio-mCherry (control groups) into VTA: (AP), −5.35; (ML), ±1.1; (DV), −8.0. After 3 weeks, all rats were subjected to a second round of surgeries with injections of CavFlexFlp into the ACC (AP), +1.6; (ML), ±0.8; (DV), −1.8 or to the OFC (AP), +3.7; (ML), ±2.1; (DV) −4.2. The behavioral experiment was conducted after 3 weeks of social separation started after the second surgery.

## Measuring of social interaction

The social interaction tests were performed after 10 minutes (control), 24 hours (baseline), or 3 weeks of social separation [47]. Initially, the tests were performed in the home cage (the data presented in Fig 1) and then in the interaction chamber (1 m × 0.5 m), which improved the video recording quality (the data presented in Figs 2, 3, 4, and S1). Rats' behavior was recorded for 10 minutes with a video camera (ImagingSource, Germany; S1 Table), and USVs were recorded with an ultrasound microphone coupled with UltraSoundGate system (Avisoft Bioacoustics, Germany) placed above the apparatus (70 cm). Both rats were placed in the chamber at approximately the same time. The animals were marked with red or green dye on their back before the test.

For assessment of c-Fos expression in the CeA, the brain tissue was collected 90 minutes from the beginning of social interaction (the rats spent the 90 minutes together). In the photomanipulation experiments, one rat from the pair was subjected to surgery (opsin delivery plus

cannula implantation). After surgery, the animals were placed back in the animal facility in single cages for 3 weeks. The partners were placed in single cages as well too. After this period, animals were reunited with a partner in a home cage to induce the expression of channelrhodopsin. Following 10-minute social interaction recording, the animals returned to the animal facility sharing their home cages. Approximately 24 hours later, rats were tested in social interaction after 10 minutes of social separation, while the activity of CeA in one rat was activated optogenetically. Light was delivered through custom-made optic cannulas (200 μm fiber, 0.39NA, Thorlabs) glued into M3 metal joints. Blue (472 nm) laser light was provided by fiber-coupled lasers (CNI), split by an optical rotary joint (Doric Lenses, 0.22 NA), and delivered through armored patch cords (Doric Lenses). Power was adjusted to obtain 10 mW at the cannula tips. The laser was triggered by an Arduino Uno microcontroller to provide 3-minute long LIGHT ON with 3-minute OFF periods (5 ms pulses, 30 Hz).

## Operant conditioning for food reward

In all the experiments in which we assessed food motivation, the animals were trained to press the lever for food before other behavioral procedures started. The behavioral training and tests were performed in Skinner boxes (Med Associates, St Albans, Vermont, USA; 35 cm × 25 cm × 25 cm); each box was equipped with light, a speaker, a grid floor, a food dispenser, and one lever located on the right side of the feeder. General training was preceded by 3-day habituation to the Skinner box. Rats were placed in the apparatus for 30 minutes, where, every 2 minutes, one sucrose pellet was delivered automatically by the food dispenser (each weighing 45 mg, Bio-Serv; S1 Table). The delivery of pellet coincided with a sound (5 seconds, 2,000 Hz at 75 dB). Additionally, the lever was present, and pressing resulted in additional reward. The subsequent training was composed of 2 stages. During the first 60 seconds, the lever was hidden. Next, the lever would appear and stay visible cyclically for 20 seconds, in 30-second intertrial intervals. Every reward delivery (one pellet per lever press) coincided with a sound (5 seconds, 2,000 Hz at 75 dB). Each training session lasted 30 minutes (35 presentations of the lever). As the first step of training, rats were trained to press the lever to receive sucrose pellets. The animals continued the first phase until they attained a stable performance: at least 70 presses a lever maintained over 3 consecutive training sessions before proceeding to the next part of training. In the second step of the training, rats had to press the lever 5 times to get the same food reward (1 pellet). To pass the second part of the training, rats had to obtain at least 250 presses a lever over 3 consecutive training sessions. The rats that passed the second part of the training continued daily sessions according to the same experimental scheme. The animals accomplished training in 10 to 12 daily-training sessions. Upon completion of the initial training, but prior to 3 weeks of social isolation, rats were subjected to 3 tests assessing the level of food motivation. The last test was used as the baseline and compared with the final test performed at the end of the experiment.

At the beginning of the Progressive Ratio (PR) test, a lever was extended into the operant chamber. The lever press was reinforced with sucrose reward, and the number of responses required for the next reward was increasing progressively with each successive reward. The reward delivery was accompanied by tone presentation (5 seconds, 2,000 Hz, 75 dB) and followed by hiding the lever for 5 seconds. During next steps, the number of required lever presses to get reward was as per Robert and Bennet implementation [48]. The exact values were 1, 2, 4, 6, 9, 12, 15, 20, 25, 32, 40, 50, 62, 77, 95, 118, 145, 178, 219, 268, 328, 402, etc. The tests lasted 30 minutes and the behavior in the final one was recorded with the video camera (ImagingSource, Germany) positioned opposite to the apparatus. The analysis was done offline.

## Experimental design for tracing experiments

After completing training for sucrose reward, all rats from this experimental group were subjected to intracranial injection of anterograde tracer PHA-L. After surgery, the animals were placed back in the animal facility in single cages for 3 weeks. Then, half of the animals were reunited with the original partners in home cages, and the other half was subjected to 30-minute testing session in Skinner's box as described above.

## Experimental design for chemogenetic experiments

In the middle of food reward training (described above), the rats had their first surgery with intracranial delivery of DREADDs. Next, rats returned to the animal facility to the pair-together home cage, and after 1 week of recovery, they went back to training in the Skinner box. After completion of the training, the rats were subjected to 24 hours of social isolation and 10 minutes of social interaction to establish the baseline social motivations. Before the second surgery (infusion of Cav-CRE or Cav-flxflp done 3 weeks after first surgery), rats had established a baseline of motivation for a food reward. Then, all animals were separated from their cagemates for 3 weeks. During social isolation, the rats were additionally exposed, 3 times per week, to 30-minute session in Skinner box (where 5 presses of the lever guaranteed one reward). After 3 weeks, animals were subjected to social interaction test. From every pair, one rat was injected with c21 diluted in NaCl (3 mg/kg, 1 mg/100 μl, intraperitoneally [IP]; S1 Table), and the second rat was injected with NaCl 30 minutes before the testing session. The c21 was administered IP, 30 minutes before the experiment began, allowing sufficient time for the c21 to take effect and remain active throughout the entire interaction, as supported by the literature [49] and our pilot studies. After 10 minutes of social interaction, the rats returned to the animal facility and were placed in home cages with their partners. In the case of groups with activation of dopaminergic pathways, animals were back to a single cage without a partner. Further isolation was necessary because after activation of dopaminergic pathways rats displayed high level of social interest disregarding social information from partners, which resulted in the increased level of fear in the partners (they performed freezing behaviors and emitted 22 kHz USVs). Additionally, 24 hours later, rats were tested for sucrose reward motivation. The animals that were injected with NaCl before the social motivation test now were given C21 diluted in NaCl (3 mg/kg, 1 mg/100 μl, IP). The rest of the animals were injected with NaCl.

## Social interaction analysis during chemogenetic neuronal projection manipulations

The data from behavioral experiments were manually scored by trained observers, blind to the experimental conditions, with frame-to-frame temporal resolution using BehaView open-source software (P. Boguszewski; S1 Table).

Behaviors were divided into 3 categories: initiation of social contacts, i.e., the active approach to a partner; maintaining social interaction, i.e., positive reactions to the partner's approach or close physical contact with a partner-crossing; and blocking social interaction where animals have avoided contacts or pushed back the partners.

**The positive social approach (initiation)** was scored whenever an animal approached a partner and performed affiliative behaviors such as nape contact, nose-to-body contact, genital investigation, following, pouncing, allogrooming, wrestling, or crawling over/under the partner's body. The new approach to partner was scored if rats walked away from the partner

(distance greater than one body length) and then went back to the partner (his head was directed towards partner's body).

**The positive reaction ratio (maintenance)** was calculated by dividing the number of positive responses to the partner's approach by the number of the partner's approaches and multiplying by 100. A reaction was considered positive when the animal directed its head to the partner's body (nose to body, genital investigation), allowed for genital investigation or allogrooming, positively responded to pouncing (e.g., rolling onto its back or engaging in pinning), and/or wrestling (the on-top and on-bottom positions alternate during social play), or allowed the partner to crawl over/under. Positive responses to the partner's approach did not include pushing or running away from the partner.

**The avoidance (blocking)** were scored when rats run away in answer to the approaches of the partner. The push-back was accounted for when animals pushed back the partners when partners tried to initiate social interaction.

Social interaction tests were additionally analyzed with Bonsai open-source software [50] to extract locomotor trajectories. The animals were identified by observing the color markings on their backs. For each animal in each frame, the central position of the marked area was determined. To assess the activity of each animal, Euclidean distance measurement was utilized.

In all categories of behaviors, animals treated with c21 were compared with conspecific partners injected with saline as well as animals from the control groups (surgery with viral vectors without DREADDs) treated with c21.

The USVs were recorded with an ultrasound microphone coupled with UltraSoundGate system (Avisoft Bioacoustics, Germany) and then analysed with RavenPro 1.5 interactive sound analysis software (Cornell Lab of Ornithology; S1 Table). The USVs were recorded throughout all the experiments. However, it is important to note that the majority of the data did not exhibit statistically significant differences. We present only the statistically significant results of USVs analyses.

## Experimental design for photomanipulation during lever pressing for food

In experiments with photomanipulation of neuronal cells in CeA activated by social interaction during lever pressing for food, the social interactions were performed in the social interaction chamber (1 m × 0.5 m). To measure positive social interaction, we scored active approaches to the partner (initiation of social interaction) and passive approaches to partner (continued interaction after approaching by partner). The rats were subject to 24 hours of social separation and 10 minutes of social interaction in order to ascertain the baseline level of positive social interaction after completion of training in the Skinner box (as described above). Social approach to the partner (positive approach to partner and positive reaction to partner's approach) and appetitive USVs were summarized and compared with social interaction after 3 weeks of social isolation. Additionally, using the PR test (described above), we established the baseline lever press level in food reward tests. Subsequently, rats were subjected to surgery (opsin delivery plus cannula implantation). After surgery, the animals were placed back in the animal facility in single cages for 3 weeks. No behavioral procedures were performed during that time. After this period, animals were subjected to either social interaction in the interaction chamber (10 minutes) or lever pressing for food (Skinner box, 30-minute session), to activate social and food cells in the CeA, respectively. Following the social interaction, animals returned to home cage, with their partners. Rats after the Skinner box session returned to the animal facility in single cages. Approximately 24 hours later, all rats were tested for motivation to obtain a food reward, while the activity of their CeA was manipulated optogenetically. On

the test day, we used the same schedule of optostimulation, delivered in 1-minute ON-laser periods for both activation and inhibition, followed by 2-minute OFF-laser periods. The number of lever presses during laser stimulation with a followed interval after stimulation was joined to one "ON–OFF batch". The first 2 adaptation minutes were considered a baseline of the test (instead of a baseline test before isolation) to exclude the possibility of uncontrolled effects resulting from 3 weeks of social isolation and connection to optic fibers. To exclude the potential long-term effects of the optostimulation of the neuronal circuits in the CeA, we analyzed only the first 2 ON–OFF batches. We chose to do so because the subsequent laser activations introduced significant fluctuations in lever pressing. For inhibition of the neurons in the CeA, we quantified all 10 ON–OFF batches as the behavioral effects were stable over the training. The difference between the long-term effects of activation and inhibition long-term effects is consistent with our earlier results with the AAV-c-Fos-ChR2(H134R)-EYFP and AAV-c-fos-NpHR-EYFP constructs, where the latter exerts less profound effects on the cell function [18]. For control, we used the same protocol as in social cells manipulations and food cells manipulations except that during the test, there was no light stimulation. Light delivery: custom-made optic cannulas (200 μm fiber, 0.39 NA, Thorlabs) were prepared before the surgeries and glued into M3 metal joints. Blue (472 nm) or yellow (589 nm) laser light was provided by fiber-coupled lasers (CNI), split by an optical rotary joint (Doric Lenses, 0.22 NA), and delivered through armored patch cords (Doric Lenses). Power was adjusted to obtain 10 mW at the cannula tips. The laser was triggered by an ArduinoUno microcontroller to provide 1-minute long LIGHT ON with 2-minute OFF periods (blue: 5 ms pulses, 30 Hz; yellow: continuous stimulation). To measure cage exploration during the PR test, trained observers, who were blind to the experimental conditions, manually scored the data from behavioral experiments. BehaView software was utilized, providing frame-to-frame temporal resolution.

## HPLC analysis

The rats were killed by decapitation immediately after the PR test with optomanipulations of the CeA circuits. The HPLC analysis was exclusively conducted on the brains of animals that underwent the PR test. The social interaction, which led to c-fos expression, occurred 1 day prior to the HPLC analysis. The data regarding the manipulation of social cells were obtained from an experiment in which specific social cells (c-fos–tagged cells) were manipulated during a PR test. Brain tissues were frozen in isopentane held on dry ice and stored at −76˚C for neurochemical analysis. Frozen brains were cut into 50 μm sections with a cryostat (−20˚C) based on the rat brain atlas of Paxinos and Watson [46]. Tissue samples were weighed and homogenized for 30 seconds in 15 vol of ice-cold 0.2 M perchloric acid, which contained dihydroxybenzylamine as an internal standard. Homogenates were then centrifuged at 26,900*g* for 8 minutes at 4˚C. Next, the supernatants were filtered through 0.45-μm pore filters and stored at −70˚C until analysis for noradrenaline (NA), dopamine (DA), 3,4-dihydroxyphenylacetic acid (DOPAC, Merck; S1 Table), homovanillic acid (HVA), 3-methoxytyramine (3-MT), using HPLC, as described by Kaneda and colleagues [51] with minor modifications [52]. The concentrations of NA, DA, DOPAC, 3-MT, HVA, MHPG, Glu, GABA, and 5-HT were calculated as ng/g of brain tissue.

## Immunohistochemistry

**Preparation of the tissue.** Two hours after the last behavioral test, rats received a lethal dose of morbital (133.3 mg/ml sodium pentobarbital, 26.7 mg/ml pentobarbital) and were transcardially perfused with ice-cold PBS (pH 7.4, Thermo Fisher Scientific; S1 Table) and 4% (wt/vol) paraformaldehyde (POCh; S1 Table) in PBS (pH 7.4). The brains were removed and

stored in the same fixative for 24 hours at 4˚C and subsequently immersed in 30% (wt/vol) sucrose at 4˚C. The brains were then fast frozen on dry ice and sectioned at 40 μm on a cryostat.

**c-Fos immunohistochemistry.** Immunofluorescence staining for c-Fos was performed on free-floating sections. The sections were washed with PBS (Thermo Fisher Scientific; S1 Table) and blocked with 3% (vol/vol) normal goat serum (NGS, Abcam; S1 Table) in PBST (0.2% Triton X-100, Polysciences; S1 Table) for 1.5 hours at room temperature. Subsequently, sections were incubated with anti-c-Fos rabbit antibody (1:1,000, Millipore; S1 Table) diluted with 5% NGS in PBST at room temperature for 24 hours. The next day, sections were rinsed with PBST, before 2-hour incubation at room temperature with a secondary antibody Alexa 594 made in rabbit (Invitrogen, 1:500; S1 Table). After several washes, the sections were mounted onto glass slides, overlaid with the Fluoromount G Medium (Merck; S1 Table), and covered with a glass coverslip.

**GFP immunohistochemistry.** GFP fluorescent staining was performed on free-floating sections. The sections were washed with PBS (Thermo Fisher Scientific; S1 Table) with 0.2% Triton X-100 (Polysciences; S1 Table), blocked with 5% (vol/vol) NGS in PBST, and incubated overnight at 4˚C with anti-GFP rabbit antibody (1:500, Millipore; S1 Table) diluted with 3% NGS (Abcam; S1 Table) in PBST. The next day, sections were rinsed with PBST before 1-hour incubation at room temperature with a secondary antibody conjugated to Alexa Fluor 488 (1:500, Invitrogen; S1 Table). After several washes, the sections were mounted onto glass slides, overlaid with the DAPI Fluoromount-G (SouthernBiotech; S1 Table), and covered with a glass coverslip.

**Th immunohistochemistry.** Th-Cre fluorescent staining was performed on free-floating sections. The sections were washed with PBS (Thermo Fisher Scientific; S1 Table) with 0.2% Triton X-100 (Polysciences; S1 Table), blocked with 5% (vol/vol) NGS (Abcam; S1 Table) in PBST, and incubated overnight at 4˚C with anti-Th mouse antibody (1:500, Millipore; S1 Table) diluted with 3% NGS in PBST. The next day, sections were rinsed with PBST before 1-hour incubation at room temperature with a secondary antibody conjugated to Alexa Fluor 488 (1:1,000, Invitrogen; S1 Table). After several washes, the sections were mounted onto glass slides, overlaid with the Fluoromount G Medium (Merck; S1 Table), and covered with a glass coverslip.

**GAD67 and mCherry immunohistochemistry.** Fluorescent immunostaining for GAD67 and mCherry were performed on free-floating sections. The sections were washed with 0.1 M TRIS (pH 7.6, Millipore; S1 Table), blocked with 5% (vol/vol) NGS (Abcam; S1 Table) in 0.1 M TRIS-BSA (0.005% BSA, Thermo Fisher Scientific; S1 Table) for 1 hour, and incubated overnight at 4˚C with anti-GAD67 rabbit antibody (1:500, Abcam; S1 Table) and anti-mCherry chicken antibody (1:500, Novus Biologicals; S1 Table) diluted in TRIS-BSA (0.005% BSA diluted in 0.1 M TRIS (pH 7.6; S1 Table). The next day, sections were rinsed with TRIS-BSA before 1-hour incubation at room temperature with a secondary antibodies conjugated to Alexa Fluor 647 (1:500, Thermo Fisher Scientific; S1 Table) and Alexa Fluor 555 (1:1,000, ThermoFisher Scientific; S1 Table). After several washes, the sections were mounted onto glass slides, overlaid with the DAPI Fluoromount-G (SouthernBiotech; S1 Table), and covered with a glass coverslip.

## Image capture and analysis

**Anterograde tracing data visualization.** The double-labelling results were analyzed with Olympus VS110 fluorescent microscope. The images were then processed with ImageJ software by experimenters who were blind to the treatments during image acquisition as well as

during cell counting. PHA-L images were merged with Venus-stained cell bodies and proximal dendrites to analyze the presence of close appositions between PHA-L and Venus-positive neurons in ROIs. The ratio of the Venus neurons receiving projections to the whole number of Venus-positive neurons was calculated.

**CeA-VTA projections visualization.** The double-labelling results were analyzed with Celldiscoverer 7 high-content phase-contrast microscope. The images were then processed with ImageJ software by experimenters who were blind to the treatments during image acquisition as well as during cell counting. The CeA inputs images were merged with GAD67 cell bodies to analyze the presence of close appositions between neuronal terminals from CeA and GAD67-positive neurons in VTA. The ratio of the GAD67-positive neurons receiving projections from CeA to the number of non-GAD67-positive neurons that receive projections was calculated.

**Verification of viral expression.** The correct expression of viruses was assessed with the aid of a Nikon Eclipse Ni-U fluorescent microscope equipped with a QImaging QICAM Fast 1394. Two objectives (20× and 10×) were used to capture samples with the aid of Image-Pro Plus 7.0.1.658 (Media Cybernetics) software.

**Statistics and data analysis.** Data are presented as mean ± SEM or as the median and quartiles using boxplots. In cases where the data distribution deviated from normality (assessed using the Shapiro–Wilk test), nonparametric tests were employed. For comparing 2 groups, unpaired $t$ tests or Mann–Whitney tests were used, while paired $t$ tests were utilized for matched pairs comparisons. To compare more than 2 groups, ANOVA with repeated measures followed by Holm–Sidak post hoc tests or Kruskall–Wallis tests followed by Dunn tests (with FDR correction) were applied. Initiations, maintenance, and blocking during the test were assessed by comparing the values to the theoretical 0 using one-sample $t$ tests or Wilcoxon signed rank test. The specific statistical tests employed for each analysis are mentioned in the figure legends. We have included the most important findings in the text. Furthermore, S2 Table provides a comprehensive presentation of all the statistical comparisons that were conducted, including detailed results. GraphPad Prism version 9 (GraphPad Software, San Diego, CA) was used for all other statistical analyses. A significance level of $p < 0.05$ was considered statistically significant. The reasons for data exclusion were as follows: (a) misplaced injection/cannulation, which resulted in inaccurate targeting or delivery of substances; (b) loss of data due to tissue damage or malfunction of the data acquisition software.

## Supporting information

**S1 Fig. Photomanipulation of the CeA social and food cells. (A)** Duration of social contacts during the 10-minute social interaction (social contacts: SI, $n = 8$; Ctrl, $n = 8$; social contact: unpaired $t$ test: t(14) = 3.233, $p = 0.006$). Cagemate rats were separated for 3 weeks and then subjected to the social interaction (SI group). The control group had the social interaction after a brief, 10-minute separation (Ctrl) **(B)** Representative image of c-Fos expression in the CeA in the SI group (lower magnification of the image shown in Fig 1C). **(C)** The experimental schematic. **(D)** Inhibition of both the food and social cells decreased lever pressing compared to baseline; in controls, there was no difference between the baseline and following phases; two-way ANOVA (time effect: F(1.460,48,17) = 12.40, $p = 0.0002$), followed by Holm–Sidak post hoc tests. The average number of lever presses per minute during the baseline period (2 minutes) and ON–OFF (3 minutes) laser periods are shown. Ctrl: Social cells: $n = 7$, Ctrl: Food cells: $n = 7$, Social cells: Inhibition: $n = 12$, Food cells: Inhibition: $n = 11$. **(E)** Time spent on exploration when the CeA cells were activated (baseline: 2 minutes, test: 6 minutes) **(F)** and inhibited (baseline: 2 minutes, test: 28 minutes). Inhibition: two-way ANOVA (time × group

effect: $F_{(3,31)} = 6.437$, $p = 0.0016$), followed by Holm–Sidak post hoc tests. No light: Social cells $n = 7$, No light: Food cells: $n = 7$, Social cells: Activation/Inhibition: $n = 7/12$, Food cells: Activation/Inhibition: $n = 9/9$. **(G)** The overall rate of lever presses per minute throughout the entire testing session when the CeA cells were activated or inhibited. **(H)** The number of social contacts and 50 kHz ultrasonic vocalizations did not differ between the groups during social interaction inducing expression of c-fos–dependent constructs. Social approaches: No light: $n = 7$, Activation/Inhibition: $n = 7/12$, USVs: No light: $n = 4$ pairs, Activation/Inhibition: $n = 5/7$ pairs. **(I)** The number of lever presses did not differ between the groups during lever pressing for food inducing expression of the c-fos–dependent constructs. No light: $n = 7$, Food cells: Activation/Inhibition: $n = 9/11$. **(J)** c-Fos expression in the CeA measured after 3-week separation (S), social interaction following separation, and lever pressing for food following separation. S: $n = 3$, social interaction: $n = 3$, lever pressing for food $n = 3$. **(K)** Both social interaction and lever pressing for food induce strong expression of the optogenetic constructs, measured as the YFP fluorescence colocalizing with the endogenous c-Fos in the social ($n = 3$) and food cells ($n = 3$), respectively. The baseline expression measured in animals with no exposure to social interaction or lever pressing for food (ctrl, $n = 3$). CeA, central amygdala; BLA, basolateral amygdala. All the data are shown as the mean ± SEM; dots represent individual data points, * $p < 0.05$, ** $p < 0.01$, *** $p < 0.001$. The data underlying this figure can be found in https://data.mendeley.com/datasets/h49vtpjm8f/3.
(TIF)

**S2 Fig. Changes in neurotransmitter levels in different brain structures after photomanipulation of the CeA social cells or food cells.** Activation (blue background) or Inhibition (yellow background) of the CeA social cells. Ctrl: $n = 4$, Social cells: Activation/Inhibition: $n = 4/6$, Food cells: Activation/Inhibition: $n = 6/6$. Black frames indicate the difference in neurotransmitter levels and their metabolites after manipulating the CeA social but not food cells. ACC; social cells-DA; Kruskal–Wallis test ($p = 0.0039$) followed by Dunn post hoc tests. Food cells-NA; one-way ANOVA (group effect: $F(2,13) = 8.344$, $p = 0.0047$) followed by Holm–Sidak post hoc tests. OFC: social cells-Glu; one-way ANOVA (group effect: $F(2,11) = 6.258$, $p = 0.0153$) followed by Holm–Sidak post hoc tests. Social cells-GABA; one-way ANOVA (group effect: $F(2,11) = 4.583$, $p = 0.0357$) followed by Holm–Sidak post hoc tests. Social cells-NA; one-way ANOVA (group effect: $F(2,11) = 13.59$, $p = 0.0011$) followed by Holm–Sidak post hoc tests. Food cells-NA; one-way ANOVA (group effect: $F(2,13) = 18.88$, $p = 0.0001$) followed by Holm–Sidak post hoc tests. mPFC: social cells-Glu; one-way ANOVA (group effect: $F(2,11) = 7.530$, $p = 0.0087$) followed by Holm–Sidak post hoc tests. Social cells-NA; one-way ANOVA (group effect: $F(2,11) = 5.326$, $p = 0.0241$) followed by Holm–Sidak post hoc tests. Food cells-NA; one-way ANOVA (group effect: $F(2,13) = 6.979$, $p = 0.0087$) followed by Holm–Sidak post hoc tests. VTA: Social cells-GABA; one-way ANOVA (group effect: $F(2,11) = 19.91$, $p = 0.0002$) followed by Holm–Sidak post hoc tests. Hipp: Food cells-Glu; one-way ANOVA (group effect: $F(2,13) = 10.42$, $p = 0.002$) followed by Holm–Sidak post hoc tests. Food cells-NA; Kruskal–Wallis test ($p = 0.0023$) followed by Dunn post hoc tests. BLA: Social cells-GABA; one-way ANOVA (group effect: $F(2,11) = 10.75$, $p = 0.0026$) followed by Holm–Sidak post hoc tests. Social cells-NA; one-way ANOVA (group effect: $F(2,11) = 10.45$, $p = 0.0029$) followed by Holm–Sidak post hoc tests. Food cells-GABA; Kruskal–Wallis test ($p = 0.0352$) followed by Dunn post hoc tests. Food cells-NA; one-way ANOVA (group effect: $F(2,13) = 17.06$, $p = 0.0002$) followed by Holm–Sidak post hoc tests. NAc social cells-GABA; Kruskal–Wallis test ($p = 0.046$) followed by Dunn post hoc tests. 5-HT; Kruskal–Wallis test ($p = 0.0251$) followed by Dunn post hoc tests. Boxplots showing median, quartiles, and the lowest and highest data points of the dependent variables, * $p < 0.05$, ** $p < 0.01$, ***

$p < 0.001$. The data underlying this figure can be found in https://data.mendeley.com/datasets/h49vtpjm8f/3.
(TIF)

**S3 Fig. Manipulating the CeA-VTA-ACC/OFC projections. (A)** Quantification of CeA projections to VTA neurons activated by social interaction ($n = 3$). **(B)** The timeline of the experiments. **(C)** Chemogenetic inhibition schematic. **(D)** Schematic of the experiment. The baseline of social interaction was measured after 24 hours of social separation. The test was performed after 3 weeks of social isolation. Before the test, one rat from the pair was injected with c21, and his partner was injected with NaCl. **(E)** Comparison to the baseline: Inhibition of the CeA-VTA projection disrupts the maintenance of social interaction. Initiation: two-way ANOVA (test × group effect: $F(5,28) = 8.376$, $p < 0.0001$) followed by Holm–Sidak post hoc tests. Maintenance: two-way ANOVA (group effect: $F(3,16) = 5.90$, $p = 0.0065$) followed by Holm–Sidak post hoc tests. **(F)** Activity after CeA-VTA inhibition. Ctrl: NaCl/c21 $n = 5/5$, CeA-VTA: NaCl/c21 $n = 5/5$. **(G)** Quantification of H3 expression in VTA Th-positive cells ($n = 2$). **(H)** Chemogenetic activation schematic. **(I)** Comparison to the baseline: The activation of the VTA-OFC dopaminergic projection decreases the initiation of social interaction by the partner rat, whereas the activation of the VTA-ACC pathway affects the maintenance of social contact by the partner rat. Initiation: two-way ANOVA (test × group effect: $F(3,16) = 5.865$, $p = 0.0067$) followed by Holm–Sidak post hoc tests. Maintenance: two-way ANOVA (test × group effect: $F(5,32) = 3.469$, $p = 0.0129$) followed by Holm–Sidak post hoc tests. **(J)** Activity after VTA-ACC and VTA-ACC activation. VTA-OFC two-way ANOVA (test: $F(1,12) = 15.35$, $p = 0.0020$), followed by Holm–Sidak post hoc tests. Ctrl: NaCl/c21 $n = 6/6$, VTA-ACC: NaCl/c21 $n = 6/6$, VTA-OFC: NaCl/c21 $n = 7/7$. Pink background: rats injected with c21 before test, light pink background: rats injected with NaCl before test. All the data are shown as the mean ± SEM; dots represent individual data points, $^* p < 0.05$, $^{**} p < 0.01$, $^{***} p < 0.001$. The data underlying this figure can be found in https://data.mendeley.com/datasets/h49vtpjm8f/3.
(TIF)

**S4 Fig. Motivation to pursue food reward after manipulating the CeA-VTA-ACC/OFC pathways. (A)** The schematic of the experiment. Rats were tested in the Progressive Ratio test to assess their motivation for obtaining food reward (sucrose pellet). The Baseline (B) measurement was taken before social separation, and the test was performed 1 day after the social interaction test. Before the test animals were injected with either c21 (pink background) or NaCl (light pink background, ctrl). **(B)** Inhibition of the CeA-VTA has no effect on the number of lever presses for food. Comparison between test (with c21 or NaCl) and baseline. Ctrl: NaCl/c21 $n = 5/5$, CeA-VTA: NaCl/c21 $n = 5/5$. **(C)** Activation of either the VTA-ACC or VTA-OFC has no effect on the number of lever presses for food. Comparison between test (with c21 or Nacl) and baseline; Ctrl: NaCl/c21 $n = 6/6$, VTA-ACC: NaCl/c21 $n = 6/6$, VTA-OFC: NaCl/c21 $n = 6/8$. All the data are shown as the mean ± SEM, and symbols represent individual data points, $^{**} p < 0.01$. The data underlying this figure can be found in https://data.mendeley.com/datasets/h49vtpjm8f/3.
(TIF)

**S5 Fig. Manipulating the ACC/OFC-CeA projections. (A)** Quantification of CeA neurons activated by lever pressing for food (left) and neurons that receive projections from ACC or OFC (right); ACC-CeA: $n = 3$, OFC-CeA: $n = 3$. **(B)** The representative images of the AAV-hSyn-DIO-{hCAR}off-{hM4Di-mCherry}on-W3SL expression in the ACC and OFC (lower magnification of the images shown in Fig 4G). **(C)** The example of image with ACC inputs in

the CeA with expression of AAV-DIO-mCherry ACC (top) and expression of Cav-Cre-GFP in the the CeA (middle) and the merged image of all the former (down). **(D)** Chemogenetic inhibition schematic. **(E)** Comparison to the baseline: (Left) Chemoinhibition of the OFC-CeA projection decreases the number of social approaches to partner. (Right) Inhibition of the ACC-CeA pathway decreases the maintenance of social interaction. Initiation: two-way ANOVA (test × group effect: $F(5,28) = 8.376$, $p < 0.0001$) followed by Holm–Sidak post hoc tests. Maintenance: two-way ANOVA (test × group effect: $F(5,28) = 3.374$, $p = 0.0165$) followed by Holm–Sidak post hoc tests. **(F)** Locomotor activity after ACC-CeA and OFC-CeA inhibition. Ctrl: NaCl/c21 $n = 5/5$, ACC-CeA: NaCl/c21 $n = 6/6$, OFC-CeA: NaCl/c21 $n = 6/6$. **(G)** Inhibition of the ACC-CeA and OFC-CeA projections decreases the number of lever presses for food. Comparison between test (with c21 or Nacl) and baseline; two-way ANOVA (test × group effect: $F(5,27) = 4.319$, $p = 0.0051$), followed by Holm–Sidak post hoc tests; Ctrl: NaCl/c21 $n = 5/5$, ACC-CeA: NaCl/c21 $n = 6/6$, OFC-CeA: NaCl/c21 $n = 6/5$. Pink background: rats injected with c21 before test, light pink background: rats injected with NaCl before test. All the data are shown as the mean ± SEM; dots represent individual data points, ** $p < 0.01$, *** $p < 0.001$. The data underlying this figure can be found in https://data.mendeley.com/datasets/h49vtpjm8f/3.
(TIF)

**S1 Table. Key resources.**
(DOCX)

**S2 Table. The results of statistical analyses.** Avoidance behavior during the inhibition of the ACC-CeA projection in response to the partner's attempts to initiate social contact (the focal rat marked in red, the partner rat marked in green). Pushing-back behavior during the inhibition of the ACC-CeA projection in response to the partner's attempts to initiate social contact (the focal rat marked in red, the partner rat marked in green).
(DOCX)

**S1 Video. Avoidance behavior during the inhibition of the ACC-CeA projection in response to the partner's attempts to initiate social contact (the focal rat marked in red, the partner rat marked in green).**
(AVI)

**S2 Video. Pushing-back behavior during the inhibition of the ACC-CeA projection in response to the partner's attempts to initiate social contact (the focal rat marked in red, the partner rat marked in green).**
(AVI)

## Acknowledgments

We are thankful to Adam Plaznik for help with the HPLC experiments and to Danuta Turzynska and Alicja Sobolewska for the invaluable technical support.

## Author Contributions

**Conceptualization:** Karolina Rojek-Sito, Ewelina Knapska.

**Formal analysis:** Karolina Rojek-Sito.

**Investigation:** Karolina Rojek-Sito, Ksenia Meyza, Karolina Ziegart-Sadowska, Kinga Nazaruk, Adam Hamed, Michał Kiełbiński, Wojciech Solecki.

**Methodology:** Karolina Rojek-Sito, Ksenia Meyza, Karolina Ziegart-Sadowska, Kinga Nazaruk, Adam Hamed, Michał Kiełbiński, Wojciech Solecki.

**Resources:** Wojciech Solecki, Ewelina Knapska.

**Visualization:** Karolina Rojek-Sito.

**Writing – original draft:** Karolina Rojek-Sito, Ksenia Meyza, Alicja Puścian, Ewelina Knapska.

**Writing – review & editing:** Karolina Rojek-Sito, Ksenia Meyza, Karolina Ziegart-Sadowska, Kinga Nazaruk, Alicja Puścian, Adam Hamed, Michał Kiełbiński, Wojciech Solecki, Ewelina Knapska.

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
