## [Editor Report · Decision Letter 0]

21 Mar 2023

Dear Dr Knapska, 

Thank you for submitting your manuscript entitled "Neuronal circuits mediating initiation and maintenance of social interaction" for consideration as a Research Article by PLOS Biology.

Your manuscript has now been evaluated by the PLOS Biology editorial staff, as well as by an academic editor with relevant expertise, and I'm writing to let you know that we would like to send your submission out for external peer review.

Once your full submission is complete, your paper will undergo a series of checks in preparation for peer review. After your manuscript has passed the checks it will be sent out for review. To provide the metadata for your submission, please Login to Editorial Manager (https://www.editorialmanager.com/pbiology) within two working days, i.e. by Mar 23 2023 11:59PM.

Kind regards,

Roli Roberts

Roland Roberts, PhD

Senior Editor

PLOS Biology

rroberts@plos.org

---

## [Decision Letter · Decision Letter 1]

17 May 2023

Dear Dr Knapska,

Thank you for your patience while your manuscript "Neuronal circuits mediating initiation and maintenance of social interaction" was peer-reviewed at PLOS Biology. It has now been evaluated by the PLOS Biology editors, an Academic Editor with relevant expertise, and by four independent reviewers. 

In light of the reviews, which you will find at the end of this email, we would like to invite you to revise the work to thoroughly address the reviewers' reports.

You will see that reviewer #1 is very positive, but wants some clarifications, including regarding the stats. Reviewer #2 is positive about the topic, but thinks that the conclusions are not well supported; this reviewer has a list of 9 “major issues,” several of which would involve substantial experimental work. Reviewer #3 is firmly positive, and appreciates the value of the dataset, but raises several significant concerns about the way the statistics have been done. Reviewer #4 is also positive, but thinks the Introduction is unclear, and has several further requests for clarification. Reviewer #4's point 2 would also involve an experiment.

IMPORTANT: I discussed the differences of opinion by inviting the reviewers to cross-comment, and then by discussing all reviews and cross-comments with the Academic Editor. The Academic Editor felt strongly aligned with the opinion of reviewer #3, and thought that while the experimental requests from reviewers #2 and #4 would undoubtedly improve the study, they will *not* be required for further consideration at PLOS Biology. However, *all* other concerns, including the analytical ones from reviewers #1 and #3, must be addressed.

Given the extent of revision needed, we cannot make a decision about publication until we have seen the revised manuscript and your response to the reviewers' comments. Your revised manuscript is likely to be sent for further evaluation by all or a subset of the reviewers.

**IMPORTANT - SUBMITTING YOUR REVISION**

*Re-submission Checklist*

*Published Peer Review*

*PLOS Data Policy*

*Blot and Gel Data Policy*

Sincerely,

Roli Roberts

Roland Roberts, PhD

Senior Editor

PLOS Biology

rroberts@plos.org

REVIEWERS' COMMENTS:

Reviewer #1:

In this manuscript, Rojek-Sito and colleagues describe an interesting set of results that provides novel insight into the neuronal circuitry regulating several aspects of social behavior in rats. The authors focus on two aspects of social interaction - initiation and maintenance - and investigate the role of the reward system in regulating these behaviors. By using a combination of activity-tagging of neurons, opsins, projection-specific chemogenetic, and functional tracing, the authors identify partially distinct neuronal circuits important for initiating and maintaining social contact. 

The findings in the manuscript are novel, and the data appear to be of high quality, with multiple controls for each circuit manipulation. It highlights the importance of breaking down complex behaviors into multiple parts to understand the underlying circuitry. I believe the paper would be of general interest to researchers in the social and behavioral field. 

I believe the paper is a good fit for PLOS Biology, and I only have some minor comments that I hope can improve the manuscript before publication: 

For the experiments inhibiting CeA-VTA in Figure 2D-G, the authors state that inhibition of the CeA-VTA disturbs the initiation and maintenance of social interaction. However, based on the figure, the change in interaction is only significantly different because the NaCl-receiving control partner rat show increased interaction. The number of approaches in the C21-treated CeA-VTA group does not seem different than the control groups, suggesting that blocking this circuit does not affect interaction but reduces maintenance and increases blocking. Could the authors elaborate on this discrepancy in the text? Could the increase in blocking be primarily caused by the increased interaction from the partner rat?

The authors observe that manipulating social circuits also affects the behavior of the partner rats. As all behaviors will at least partially depend on what the partner rat is doing, it may be useful to show the data as paired interactions instead of individual animals to more clearly show which of the animal is driving the change in behavior. However, this would not be essential to support the paper's conclusion. 

Although the statistics in clearly stated in the figure, a statistics and data analysis section is missing from the method. A statistic section should be included to show if tests for normality were performed, if there were any exclusion criteria, and how significance was calculated. 

Some of the investigated circuits appear more specific for social behavior, whereas others may more generally regulate reward as their manipulation also affects food reward. It would be helpful to more clearly state in the discussion which circuits are social-specific and some discussion on the distinction between more general versus specific circuit functions. 

Reviewer #2:

The manuscript titled "Neuronal circuits mediating initiation and maintenance of social interaction" focuses on the circuit mechanisms underlying the initiation and maintenance of social interactions, utilizing virus tracing, DREADDs, and behavior. The author has identified two multisynaptic closed pathways that originate from the CeA and return to it, which are involved in initiating and maintaining social interactions. Given that social ability is crucial for the survival of social species, exploring this topic is deemed worthy. However, some of the findings from the study are not yet sufficiently robust and convincing to fully support the primary claims. In order to strengthen the "causal link" between the circuits and social interaction behavior, several concerns must be addressed.

Major issue:

1. The author began to investigate the social cells in the CeA using Fos-activated virus expression. However, they did not study the projections of these social cells to the VTA. They used a retrograde Cav-cre and DIO-Gi strategy to label the projection neurons from the CeA to the VTA. We did not know how many of these cells were social cells. When studying the projection from the ACC or OFC to the CeA. They analyzed the ratio of the projection of ACC or OFC to the social cell in the CeA again. There is no direct functional evidence to show the circuits involved in social initiation and maintenance.

2. The author claimed that the projection neurons from the CeA to the VTA are GABAergic. However, there is a lack of both morphological and electrophysiological evidence to support it.

3. The author used THcre mice (DA neurons) to investigate the function of the VTA-ACC and VTA-OFC pathways. Based on the results shown in Figure 2, the projection neuron of CeA sends its axon to VTA GABAergic neurons.

4. The author primarily utilized the DREADDs Gi strategy to verify the circuit's involvement in social interaction. It is better to confirm it using the DREADDs Gq strategy. For initiation and maintenance, real-time optogenetics may also be necessary.

5. The dopamine receptors are G-protein-coupled receptors, which means they do not directly modulate the neuronal activities of ACC and OFC neurons. The effect does rely on the type of dopamine receptor. What is the latency of VTA-ACC Gq experiments? It is better to provide evidence demonstrating that the neurons in the ACC and OFC are indeed activated by C21.

6. To determine whether the social and food cells overlap, the author could induce the expression of social and food cells at different stages using two different colors and count the number of cells.

7. In Figure 2D, why do the No. approaches of the CeA-VTA:NaCl group and Control:NaCl group show a significant difference? It is the same as in Figure 3G.

8. The evidence of CeA predominantly projecting to VTA GABAergic neurons in Figure 2H-I is not strong enough.

9. Do the same social cells in the CeA receive projections from both the ACC and OFC, or are they two distinct groups?

Minor issue:

1. In Figure 1B, what is the total or average duration of social contacts?

2. Figure 1C lacks an image at low magnification to clearly show that the area is indeed CeA. Meanwhile, it is better to co-stain cFos with NeuN. What is the cell type of cFos-positive cells?

3. In Figure 3B, it appears that there are still some mCherry-positive signals that do not co-express with TH-positive cells. It is better to calculate the ratio of double positive cells to mCherry cells.

4. In Figure 4G, there is a lack of low magnification images to show the infection area of ACC and OFC.

5. Figure 3B shows that the TH+ cells are actually Cre-positive cells. It still needs to be stained with TH or DA to confirm that they are DA cells.

Reviewer #3:

In this manuscript, Rojek-Sito and colleagues address a very interesting and important question, how social interactions are instantiated at the level of multiple neural circuits, performing a very elegant and detailed set of loss and gain of function experiments in specific circuits. The experiments and methodological approaches to address this important question are exceptional, and the dataset generated is clean and robust. This is a heroic amount of work, very exhaustive which provides an extremely valuable dataset. From these experiments and analyses, the authors claim to identify partially different circuits that are involved in different aspects of social interactions.

I would like to thank the authors to have devoted so many efforts in dissecting specific aspects of social behavior. These are not easy questions to tackle, and are not usually addressed, favoring simplistic approaches to social interactions, which are not good for the field. I thus really appreciate the efforts that the authors have dedicated to make sense of this complex data set. 

In conclusion, this is a very valuable piece of work, I have no concerns on how experiments are performed and the techniques chosen to address, sequentially, each of their questions. This work is really beautiful, although it was difficult for me to fully understand all claims of the work in the current form. Nevertheless, I enjoyed reading it and made me learn and question many aspects of the social brain. I am convinced that this work is going to be important, I have however one main concern, before I can provide my full support for acceptance for publication in Plos Biology (there are several minor issues that I will list at the end). My main concern is on how the statistical analysis are performed, which might be affecting the conclusions that the authors obtain. 

The authors separate 3 aspects of a social interaction: (1) Initiation, (2) Maintenance and (3) Avoidance. Initiation is clear. Maintenance, for the authors are the positive interactions addressed to maintain the interaction after it is initiated. I see avoidance as the inverse of the measures of maintenance, to stop an interaction after it is initiate, thus could be considered as (the negative of) maintenance. This avoidance is a very meaningful measure, but is not taken into account to draw the conclusions (and omitted in some graphs and graphical abstract, for example). 

However, this is not my main concern. The first big issue I have is how statistics were performed to identify effects on initiation vs maintenance. It took me a long time to understand that, for example in the DREADDs experiments, the saline and c21 animals were interacting within the same pair. However, the statistics performed are done considering that they are independent samples. There is not a unique manner to analyze social interactions, I am very open to discuss with the authors (or editor) on the positive and negative aspects of each statistical analyses, but in this case, and as the authors are well aware and recognize in their discussions, social interactions are bidirectional, so change in the behavior of one animal, will affect the behavior of the other. I think thus that it is not correct to treat these groups as independent samples. 

In order to find changes in the social interactions with their circuits manipulations, authors make an extra analysis where they compare the behavioral variables within each animal in a baseline vs a test session. However, the baseline was performed after a short isolation and test after 3 weeks of isolation. I do not think these conditions are comparable and the best way to question on the effects of their circuit manipulations. In my opinion, the most significant comparison would be paired samples t-tests between each animal of a pair on the test day. This will reveal which aspects of the social interaction are affected. 

Nevertheless, all the data provided from the control/baseline groups are very valuable and useful to see. With these graphs we understand that there are no previous differences between experimental groups, and can observe inter-individual variability, but I would not compare these values with the ones after 3w of social isolation. These data should be provided maybe in the supplementary materials, is useful to see, but omitted from the main figures, which might help also to simplify the main message.

I realize that performing paired t-test in the test days of the experimental manipulations might provide different results than the ones obtained so far, although I have been checking in some experiments with the raw data that authors gently made already public, and most of the main conclusions do hold. In some cases (ACC experiments) it will change a bit, but I still think it is more accurate this way of doing the statistics.

The other big concern I have, which evolves from the new way of doing statistics I suggest, is how the maintenance variable is currently calculated. As it is now, this variable is the sum of 2 variables: positive interactions ratio + crossings. My concern is specially on the calculation of the passive positive interactions normalizing by the number of initiations of the other animal. I understand the logic of this, but in this manner, in some cases, the effects seen in the maintenance are a consequence of the differences observed in the initiation of social interaction by the other animal. I would think that it would dissect better the contribution of each circuit if the variables (initiation vs maintenance) would not depend on each other. So I would propose to revisit the maintenance data with the number of passive interactions of the test only, not making it relative to the initiation variable, nor adding the crossings (as we can not disambiguate which animal of both perform the crossing and it adds the same amount equally to both animals of the pair).

I would understand that a normalization to the frequency of initiation events of the partner would be performed if the duration of the passive approaches would be studied, but not so much in the frequency. Indeed, when the authors present the number of avoiding and pushing events, they are not normalizing by the number of initiations of the partner. I think that this new way of analyzing the data might provide cleaner differential effects of the maintenance variables (and circuits) which at the moment I fear are contaminated by how many initiations the partner does. 

These are the main concerns that I have at the moment. I am open to hear the opinion of the authors of these new analysis, although I think they might help to better delineate the differential contribution of each circuit in either initiation or maintenance. As I see it in this moment, the null effects that support the conclusions, are not strongly null, and I fear it is due to these statical/data definition used in this current version.

Then, I will list some of my minor concerns, that I hope that also help to make this manuscript even better.

MINOR CONCERNS:

- line 75: there is a missing "." just after the reference and Nevertheless

- Fig 1B: where there any 22kH USVs in these first interactions after social isolation? If not, it would be interesting to comment this in the text, to support the statement that no agonistic interactions were observed, for example in line 95.

- LINE 109: just a style suggestion, as it is now, it is uncertain if "the previous work" refers to someone else's work of the first results that you have provided so far. Once you read the full sentence one can understand that it refers to previous literature. Maybe change the starting of this sentence to something like Previous work has shown.

- Fig S1: Figure legend (G) says the following "(G) Inhibition of both the food and social cells decreased lever pressing compared to baseline, in controls there was no difference between the baseline and following phases; two-way ANOVA (time effect: F(1.460,48,17) = 12.40, p=0.0002), followed by Holm-Sidak post-hoc tests." Does this F and p value refer to the all 4 groups? Does this mean that there is no interaction between the time and the experimental group? Because then it would mean that the decrease in the on-off period is also seen in the controls, meaning that the number of lever presses decreases over testing, which is expected. I understand that posthoc comparisons provide only significant differences in the "light" groups, however this posthoc comparisons are only justified if in the 2way ANOVA you have an interaction with time and experimental groups. In general, I find it confusing not to have all the statistical descriptives so we can see if the within or between subjects factors or the interactions are significant.. Could you please clarify me the rationale of only providing part of the statistical analysis? Maybe you are only providing the statistics that are significant to simplify? I would be ok with that, but maybe it would be good to state it in the statistics section of the methods.

- Fig S2: at the end of the figure legend is written that Avg and sem is shown, however, in this figure, boxplots are used, which is indeed more informative, and correct in case of not normal distribution. Please correct, and indicate what the centroid line and whiskers indicate.

- Regarding the Figures representing % [test-ctrl] in Initiation /Maintenance (for example fig 2G, fig3H, fig 4K). Why not to include here the Blocking results? (This comment will only hold if authors decide not to change the statistics as I suggested when articulating my main concerns). These are very helpful for providing a simplified view of the results. However, I wonder on whether the adequate statistics here would be to compare each group for each circuit against the 0 (meaning no change from baseline). As it is now, for example in fig3H the message is that VTA-ACC activation impacts more the initiation than the maintenance, and that difference is not significant in VTA-OFC. I wonder if the same message holds when asking if each circuit changes its own baseline (by differing from 0). 

- While considering the initiation/maintenance/blocking results, I think is a good manner to explain different parts of social interaction, although they will be dependent on each other. For example, if one animal initiates many contacts, it might elicit more blockings by the partner. Did you explore correlations between these variables and how they increase or weaken for each circuit manipulations? For example, if inhibiting one circuit induces an animal to initiate less contacts, it might be that the partner will initiate more, and then the experimental animal will block more.

- line 898: "Activation of both pathways results in active blocking of social contact by the partner". Could you please add at the end of this sentence "by the partner of the treated animals" or similar? I only realized that the slaine and the c21 animals were in the same pair after reading the full manuscript the first time and reaching the methods. It would help if this was stated at the beginning of the manuscript. Also, authors should think if they want to reduce the statistical comparisons to those that are relevant (here, for example, the activated circuit with c21 against their own control) instead of comparing everything with everything. If authors want to compare everything with everything, maybe corrections for multiple comparisons should be performed.

- In general, please indicate in the figures whether the circuits are activated or inhibited, this would help readers to follow your results (for example fig s4)

- Fig S4: I am confused of the statistical comparisons used to obtain the conclusions regarding the social interactions. For example, in FifS4 D it is stated that: " Activation of VTA-ACC dopaminergic projection increases the initiation of social interaction. " However there are many more comparisons there that are statistically significant, also regarding the VTA-OFC. Could you please clarify? If the focus is on the within-subjects comparisons, then repeated measures anova should be used (baseline and test as within-subjects and treatment as between subjects). However, I am not sure if this is the best comparison to extract conclusions, as the values of a social interaction after 24h or after 3w isolation are going to differ. The baseline values are useful to see that prior manipulations the values of social interaction are similar, and to understand individual variability within each experimental group, but I would not take them as the statistical reference for the circuit manipulations.

- If within subjects (baseline/test) statistics are decided, then reorganize the graphs as in Fig S6B, where the 2 groups to compare are next to each other.

- Do the authors have USV data from the experiments of the circuits manipulations? It is only presented in Fig 1. If only USV data is available for the 1st figure, please specify this in the methods section

- In methods section, it is described that the behavioral data of Figure 1 was obtained studying interactions in the home-cage, and then, an open-field type cage was used, which improved tracking. Later, it is written that c-fos induced expression of channelrhodopsin was performed in a home cage, I imagine to reduce the expression of cfos while exposed to a novel environment. This is consistent for Figure 1, but where did the behavioral tests happened in Figure 4? Could you please clarify?

- Line 485: it is stated that Bonsai was used to extract locomotor trajectories in social interaction tests. It would be nice that the authors explain here how did they created the workflow that allowed them to differentiate the trajectories of each individual animal (I assume using color), this is not trivial, and a short sentence describing how they performed this, would be useful. Moreover, did they use Bonsai to measure locomotor activity in the Food-progressive ratio test? I see that time of exploration is presented in supplementary figures, but not sure how this was tracked.

- Line 497: Positive reaction ratio (maintenance). I was very confused the first time I read the manuscript, as only when I reached the methods definition I understood that this was a ratio. If at the end authors decide to include this ratio as their measure of maintenance of social interactions, it would be advisable that the write that is a ratio in the y axis legend of the figures. However, I thing that if the maintenance values are divided by the number of interactions, then they will be influenced by the INITIATION results.

- Line 561: I think there is an extra "." to be removed between cryostat (-20ºC) and Based on the rat brain atlas

- Line 557: it is written that HPLC was performed in brains of animals after the progressive ratio test, however, they have performed them in animals after the social interaction test, too. Also, it is written that brains were cut into sections with a cryostat (please specify thickness), but not how the different brain areas were isolated for further neurochemical analyses.

- Regarding these results, plotted in Figure S2, it would be advisable that the difference between ACC, OFC and PFC, as for my understanding both ACC and OFC could be also considered as prefrontal cortex. Is PFC referring to medial prefrontral cortex specifically (IL, Prelimbic)

Reviewer #4:

The authors of Rojek-Sito et al study address an important question of how different neural circuits underly initiation and maintenance of social interactions. While this topic is highly relevant and currently underexplored, there are several weaknesses in the way some experiments are conducted and the data is interpreted. The authors will need to address these points in order to ensure that the data presented support their main conclusions.

Major:

1) The Introduction is rather poorly written and does not reflect the importance of the questions addressed in the study. In difference to many other studies that investigate social interactions in general, this study is focusing on functionally differentiating between two stages of social interaction - initiation and maintenance. Before addressing the putative involvement of the neural circuits, I would suggest to elaborate on the evolutionary importance of the two stages of social interaction. Furthermore, Line 81-83, the authors state: "Further, we explored the neuronal circuits underlying initiation, maintenance, and responses to partners' attempts to initiate contact, all of which have to be synchronized to build a successful social interaction" - in this study, no synchronization has been measured, please rephrase. Additionally, on lines 72-73 the authors mention that the CeA modulates emotion discrimination during social interaction but do not explain what that means and why it is relevant for the hypothesis. Please elaborate. 

2) The authors perform 3 weeks social separation prior to the test. While they are citing a paper (Gorlova et al, 2018) demonstrating that 3-weeks isolation does not lead to depressive-like disorders in rats, such a procedure certainly induces stress. Given that central amygdala is also involved in stress-related behavior and is affected by chronic stress, the authors will need to demonstrate that the increased CFos expression they observe is not related to the increased stress level. The experiment that should be conducted is following: social separation for 3 weeks after which the animals should be put in the context without social interaction - CFos expression level should be then compared with CFos expression during the social interaction. Also, the authors should cite the effects of stress throughout the text, in the introduction and discussion.

3) While the results presented in Figure 1 related to social and food cells are interesting, I am somewhat confused regarding the authors' interpretation about the overlap between the two cell populations. As far understood, the activation is done in two groups, but then in inhibition experiments, the authors claim to inhibit both social and food cells simultaneously. How was that done? Please clarify. Did the authors conducted the social and food experiments in two different groups and looked into the Cfos expression in CeA? Does the expression pattern look similar in terms of C-Fos cell density and anatomical location? This might offer further insight into the potential overlap between the social and food cells, or lack thereof. 

4) In the Methods section, line 541-545, the authors state: "To exclude the potential long-term effects of the optostimulation of the neuronal circuits in the CeA, we analyzed only the first two ON-OFF batches. We chose to do so because the subsequent laser activations introduced significant fluctuations in lever pressing. For inhibition of the neurons in the CeA we quantified all 10 ON-OFF batches as the behavioral effects were stable over the training" The authors refer to their previous study where they show that the effect of activating opsin is less reliable. I found this explanation somewhat confusing and would like to see the data 3-10 ON-OFF batches for the activating condition. 

5) The experiments with measuring neurotransmitter release are interesting, but the reasoning behind this type of experiments as well as rationale to focus on specific pathway are insufficiently explained. Please elaborate on your rationale to focus on the decrease of GABA in VTA and not, for example on the decrease of Glu or NA in PFC. Also, the authors did not perform any measurements in Nucleus Accumbens, which also receives CeA inputs, please clarify. 

6) Dopaminergic VTA-ACC - the initiation effect for VTA-ACC is similar to the maintenance effect of VTA-OFC (Figure 3 F). This was not mentioned/discussed. Further, in H - in the legend inhibition is mentioned although the figure appears to contain only activation - please clarify. In addition, no reference to this result (graph H) is given in the text. 

Minor

Highlight 2: please clarify

Line 107: "…CeA cells causally involved…", so far the experiment does not delve into activating or inhibiting these cells, so the authors should use "associated" instead.

Lines 130-131: Sentence is confusing, please clarify.

Lines 134-136: The rationale behind the next experiments are explained here, but in a rushed way. Please elaborate.

Lines 154-155: Please elaborate how (with double labeling?)

Lines 156-158: The authors want here to suggest that the CeA output to VTA inhibit the tonic inhibition of dopaminergic neurons via gabaergic interneurons, but the text is confusing. It should be clarified.

Lines 212-214: Sentence is confusing and poorly written. Please correct.

Line 229: …whole role…should be: which role 

Lines 252-254: Please elaborate here and cite the studies.

Lines 259-261: Sentence is confusing, please clarify.

---

## [Decision Letter · Decision Letter 2]

4 Sep 2023

Dear Dr Knapska,

Thank you for your patience while we considered your revised manuscript "Neuronal circuits mediating initiation and maintenance of social interaction" for publication as a Research Article at PLOS Biology. This revised version of your manuscript has been evaluated by the PLOS Biology editors, the Academic Editor, and two of the original reviewers.

Based on the reviews and on our Academic Editor's assessment of your revision, we are likely to accept this manuscript for publication, provided you satisfactorily address the remaining very minor point raised by reviewer #3, and the following data and other policy-related requests.

IMPORTANT - Please address the following:

a) Please change the Title to include an active verb and the methodology. We suggest: "Optogenetic and chemogenetic approaches reveal differences in neuronal circuits that mediate initiation and maintenance of social interaction"

b) Please include the name of the study species in the Abstract.

c) Please address the remaining point raised by reviewer #3.

d) Please address my Data Policy requests below; specifically, we need you to supply the numerical values underlying Figs 1BDGHK, 2DEFGI, 3EFGH, 4BCHIJK, S1ADEFGHIJK, S2, S3AEFGIJ, S4BC, S5AEFG, either as a supplementary data file or as a permanent DOI’d deposition. I note that you already have an associated Mendeley deposition (https://data.mendeley.com/datasets/h49vtpjm8f/2), and while individual tabs are mostly labelled with the names of Figures, it's unclear how these relate to specific Figure panels. Please could you clarify this?

e) Please cite the location of the data clearly in all relevant main and supplementary Figure legends, e.g. “The data underlying this Figure can be found in https://data.mendeley.com/datasets/h49vtpjm8f/2”

f) Please make any custom code available, either as a supplementary file or as part of your Mendeley deposition.

We expect to receive your revised manuscript within two weeks. 

*Published Peer Review History*

*Press*

Sincerely,

Roli Roberts

Roland Roberts, PhD

Senior Editor,

rroberts@plos.org,

PLOS Biology

DATA POLICY:

Regardless of the method selected, please ensure that you provide the individual numerical values that underlie the summary data displayed in the following figure panels as they are essential for readers to assess your analysis and to reproduce it: Figs 1BDGHK, 2DEFGI, 3EFGH, 4BCHIJK, S1ADEFGHIJK, S2, S3AEFGIJ, S4BC, S5AEFG. NOTE: the numerical data provided should include all replicates AND the way in which the plotted mean and errors were derived (it should not present only the mean/average values).

CODE POLICY

Per journal policy, as the code that you have generated is important to support the conclusions of your manuscript, we require that you make it available without restrictions upon publication. Please ensure that the code is sufficiently well documented and reusable, and that your Data Statement in the Editorial Manager submission system accurately describes where your code can be found.

SPECIES INDICATED IN THE ABSTRACT? 

- Please note that per journal policy, the model system/species studied should be clearly stated in the abstract of your manuscript. 

DATA NOT SHOWN?

REVIEWERS' COMMENTS:

Reviewer #3:

[identifies herself as Cristina Marquez]

 I would like to thank the authors for the careful and thorough revision that they have done in the present form of the manuscript. All my concerns were addressed and this version is much more direct and understandable.

Please revise the sentence in Line 124: " we targeted halorhodopsin" do you mean we expressed? Transduced? 

Reviewer #4:

[identifies herself as Sanja Mikulovic]

The authors have satisfactorily responded to all my concerns. I congratulate them to this diligent work which will certainly make a strong impact on the field of rodent social neuroscience.

---

## [Editor Report · Decision Letter 3]

20 Sep 2023

Dear Dr Knapska,

Thank you for the submission of your revised Research Article "Optogenetic and chemogenetic approaches reveal differences in neuronal circuits that mediate initiation and maintenance of social interaction" for publication in PLOS Biology. On behalf of my colleagues and the Academic Editor, Matthew Rushworth, I'm pleased to say that we can in principle accept your manuscript for publication, provided you address any remaining formatting and reporting issues. These will be detailed in an email you should receive within 2-3 business days from our colleagues in the journal operations team; no action is required from you until then. Please note that we will not be able to formally accept your manuscript and schedule it for publication until you have completed any requested changes.

Sincerely,

Roli Roberts

Senior Editor

PLOS Biology

rroberts@plos.org